# Genetic Diversity and Population Structure of Common Bean (*Phaseolus vulgaris* L.) Landraces in the Lazio Region of Italy

**DOI:** 10.3390/plants12040744

**Published:** 2023-02-07

**Authors:** Giulio Catarcione, Anna Rita Paolacci, Enrica Alicandri, Elena Gramiccia, Paola Taviani, Roberto Rea, Maria Teresa Costanza, Gabriella De Lorenzis, Guglielmo Puccio, Francesco Mercati, Mario Ciaffi

**Affiliations:** 1DIBAF, Università degli Studi della Tuscia, Via San Camillo de Lellis, 01100 Viterbo, Italy; 2ARSIAL, Via Rodolfo Lanciani 38, 00162 Roma, Italy; 3DISAA, Università degli Studi di Milano, Via Celoria 2, 20133 Milano, Italy; 4IBBR, CNR, Corso Calatafimi 414, 90129 Palermo, Italy

**Keywords:** genetic resources, landraces, population structure, seed storage proteins, SSR markers

## Abstract

Common bean cultivation has historically been a typical component of rural economies in Italy, particularly in mountainous and hilly zones along the Apennine ridge of the central and southern regions, where the production is focused on local landraces cultivated by small-scale farmers using low-input production systems. Such landraces are at risk of genetic erosion because of the recent socioeconomic changes in rural communities. One hundred fourteen accessions belonging to 66 landraces still being grown in the Lazio region were characterized using a multidisciplinary approach. This approach included morphological (seed traits), biochemical (phaseolin and phytohemagglutinin patterns), and molecular (microsatellite loci) analyses to investigate their genetic variation, structure, and distinctiveness, which will be essential for the implementation of adequate ex situ and in situ conservation strategies. Another objective of this study was to determine the original gene pool (Andean and Mesoamerican) of the investigated landraces and to evaluate the cross-hybridization events between the two ancestral gene pools in the *P. vulgaris* germplasm in the Lazio region. Molecular analyses on 456 samples (four for each of the 114 accessions) revealed that the *P. vulgaris* germplasm in the Lazio region exhibited a high level of genetic diversity (He = 0.622) and that the Mesoamerican and Andean gene pools were clearly differentiated, with the Andean gene pool prevailing (77%) and 12% of landraces representing putative hybrids between the two gene pools. A model-based cluster analysis based on the molecular markers highlighted three main groups in agreement with the phaseolin patterns and growth habit of landraces. The combined utilisation of morphological, biochemical, and molecular data allowed for the differentiation of all landraces and the resolution of certain instances of homonymy and synonymy. Furthermore, although a high level of homozygosity was found across all landraces, 32 of the 66 examined (49%) exhibited genetic variability, indicating that the analysis based on a single or few plants per landrace, as usually carried out, may provide incomplete information.

## 1. Introduction

The common bean (*Phaseolus vulgaris* L., 2n = 2x = 22) is the most important legume for human nutrition worldwide since it is a valuable source of high-quality proteins, carbohydrates, vitamins, minerals, fibres, phytonutrients (mainly phytosterols and flavonoids), and antioxidants [1,2,3,4]. Most of these compounds have significant positive impacts on human health. Therefore, the common bean can be considered a potential functional food. Belonging to the *Fabaceae* family, the common bean also plays a significant role in sustainable agriculture due to its ability to fix atmospheric nitrogen, so reducing the need for fertilizer applications.

Nearly half of the world’s dry bean production is provided by the American continent, with the main producers being Brazil, the USA, Mexico, and Central American countries, whereas China, India, and Myanmar are the leading Asiatic producers [5]. As a healthy and cheap potential alternative to animal proteins, the common bean is also essential to the nutrition of developing countries in Africa. In Europe, cultivation is largely focused on countries surrounding the Mediterranean basin, including the Iberian Peninsula, Italy, Greece, and the Balkan regions. In Italy, the common bean is currently the most widely grown legume [6], with an annual dry bean yield of about 12,000 tonnes and more than 6400 hectares cultivated [5], although this production is not sufficient to meet the country’s demand [6].

The wild ancestor of *P. vulgaris* probably evolved in the Mesoamerican region, most likely in Mexico [7,8,9], although an Andean origin of the common bean cannot be ruled out [10]. Before domestication, wild bean forms, widely distributed from Northern Mexico to north-western Argentina, diverged about 111,000 years ago [11] into two major geographical gene pools, namely the Mesoamerican and the Andean [9,12]. Common bean wild forms comprise an additional third gene pool represented by populations that grow in a restricted region between Northern Peru and Ecuador [13]. This gene pool has only been documented for wild populations, and no domesticated forms have ever been identified. Previously, these wild populations were thought to be the possible ancestors of *P. vulgaris* [13,14]. However, according to the hypothesis of the Mesoamerican origin of the common bean, recent findings indicate that both of the wild gene pools from South America arose from independent migrations of Mesoamerican wild populations before the domestication of *P. vulgaris* (reviewed in [9]).

Domestications from wild beans took place separately in Mesoamerica and Andean Southern America and resulted in two main different gene pools within the cultivated forms. The existence of distinct domestication processes has been well established, initially by morphological and agronomic traits [15], biochemical markers [16,17], and more recently, molecular markers covering wider genomic regions [18,19,20,21,22,23,24,25,26,27,28,29,30,31].

Bean dissemination from the Americas to Europe began in the early 16th century when Portuguese and Spanish sailors and traders brought bean seeds from both centres of domestication to their respective homelands [32,33]. The spread of the crop across continents has led to genetic erosion, as only a portion of the ancestral gene pools has been transferred into Europe. The bottleneck effect was partially mitigated in Europe by the extensive gene flow between the Andean and Mesoamerican gene pools in this continent, which was greater than that observed in the Americas [24,34]. The European continent is now considered a secondary diversification centre for *P. vulgaris* because new forms, better adapted to the environmental and pedological conditions in the different geographical areas of Europe, arose from the initial recombination events between the Andean and Mesoamerican gene pools [24,34]. Indeed, the adaptation to changing environmental conditions, biotic or abiotic stress, and deliberate or unintentional selection by farmers may have exerted a significant impact on the evolution of the European common bean, resulting in the emergence of a great number of landraces [34].

Landraces are distinct but heterogeneous populations strongly adapted to local environmental conditions: they are closely linked to the traditions and cultures of the people who have developed and grown them for many years [35]. They are mostly cultivated in marginal areas with low-input production systems and possess valuable adaptation traits to various stressful environments. Although landraces are an important component of agrobiodiversity, most of them are now in danger of genetic erosion since they are cultivated by old farmers and are gradually being substituted by modern cultivars [35]. Currently, breeding programs are required to cope with climate changes as well as to meet the demands of new sectors like organic farming for which there is no specifically developed variety. From this perspective, landraces, exhibiting several specific ecological adaptations, are considered suitable for the development of useful materials for sustainable agriculture. Furthermore, consumers tend to be attracted to food products that are differentiated from others by traits, qualities, or geographical origin. Therefore researchers, consumers, and policymakers are paying increased attention to traditional common bean landraces. The key policy objective for the protection and utilization of agrobiodiversity should be based on evaluating the existing diversity, not only between but also within landraces [36]. This endeavour is crucial for the implementation of appropriate ex situ and in situ (on-farm) conservation strategies and is a prerequisite for the establishment of feasible breeding programs [36].

In Italy, common bean cultivation has been a traditional component of rural economies, particularly in mountainous and hilly areas along the Apennine ridge of the southern and central regions, where a wide diversity of bean landraces has been grown for generations [37,38]. The germplasm from several Italian regions such as Sardinia, Campania, Calabria, and Sicily has been characterized using morphological traits and biochemical and molecular markers [39,40,41,42,43,44]. However, these studies aimed to mainly evaluate the genetic relationships between accessions; hence, sampling was not exhaustive because it was limited to a single plant per landrace. However, it is well known that independent analyses of many individuals are required to adequately assess the within-landrace genetic diversity.

In the present study, a large *P. vulgaris* collection from the Lazio region (Central Italy) was characterized for morphological traits and biochemical and molecular markers. This collection included several landraces faced with the danger of extinction, and well-known landraces appreciated by the consumers for their organoleptic and nutritional properties [e.g., “Cannellino di Atina”, the only common bean landrace with the EC PDO (European Commission Protected Designation of Origin) mark] and/or with historical traditions or religious connections (e.g., “Fagiolo del Purgatorio”), currently poorly investigated. A multidisciplinary approach to the common bean landrace collection of the Lazio region was used to (i) evaluate the inter- and intrapopulation diversity and genetic structure, (ii) investigate the genetic relationships between landraces to also identify possible homonymy and synonymy cases, (iii) identify the original gene pool (Andean and Mesoamerican) of the collection studied, and (iv) evaluate the possible cross-hybridization events between the two gene pools of origin in the *P. vulgaris* germplasm in the Lazio region.

## 2. Material and Methods

### 2.1. Plant Material

Since the beginning of the 2000s, extensive monitoring of bean landraces has been conducted by ARSIAL (Lazio Region Agency for Agricultural Development and Innovation) in the entire Lazio region and conserved ex situ at the ARSIAL repository, within the framework of the Regional Law no. 15 dated 1 March 2000 “*Protection of autochthonous genetic resources of agricultural interest*”. According to historical information, only seed lots grown on a single farm for many generations (at least 50 years) were collected and each lot was considered a single accession. Information about the farm, use, local names, geographical data (latitude, longitude, and altitude), traditions, and social context were recorded for each seed lot included in the ARSIAL collection. One hundred fourteen accessions, grouped into 66 putative different landraces based on the local names attributed by the farmers, were collected in farms located in the five provinces into which the Lazio region is divided: 33 each from the province of Rieti (RI) and Frosinone (FR), 26 from the province of Viterbo (VT), 18 from the province of Rome (RM) and 4 from the province of Latina (LT) (Appendix A). The map indicating the collection sites of the accessions is available in Appendix A. Two accessions from Central America (BAT93 and G1287) and two from South America (Jalo EEP558 and Midas), obtained from Centro International de Agricoltura Tropical (CIAT, Columbia), were used as standard genotypes for Mesoamerican (the first two) and Andean (the others) gene pools, respectively, in the genetic analyses [42].

### 2.2. Morpho-Phenotypic Seed Analysis

Seed traits were measured in a random sample of 20 seeds for each accession investigated according to the International Board for Plant Genetic Resources (IBPGR) [45], using the most relevant descriptors for *P. vulgaris* [46]. Among quantitative traits, seed height (H, mm) is measured as the longest distance perpendicular to the length, seed length (L, mm), as the longest distance across the seed parallel to the hilum, and the elongation index, as the ratio between length and height (L/H) were recorded. Four qualitative seed descriptors were also recorded: coat pattern (SCP), coat darker colour (CSCD), coat lighter colour (CSCL), and shape (SSH). According to Angioi et al. [39], a last qualitative character named “prevalent” was established for the seed coat colour, with three possible classes: (1) lighter colour as the background and darker colour as stripes; (2) darker colour as the background and lighter colour as stripes; and (3) darker colour and lighter colour equally distributed. Combining the five qualitative seed traits, each accession was identified by a numerical code that represents the so-called “seed morphotype”. In addition, the weight of 100 seeds (g) was measured by using two random samples of seeds for each accession.

### 2.3. Growth Habit

The plant growth habit of the 114 accessions was determined directly in the field by visiting the farms where they are cultivated. The accessions were classified as determinate (type I growth habit) or indeterminate [47].

### 2.4. Phaseolin (PHAS) and Phytohemagglutinin (PHA) Analysis

Four seeds of each accession were used to determine the PHAS and PHA protein patterns. After removing the coat, each seed was cut into two parts and the half without the embryo was finely ground in liquid nitrogen before PHAS and PHA extraction according to Limongelli et al. [48]. In particular, PHAS was extracted by suspending about 20–40 mg of flour for 30 min in 0.5 M NaCl (1/10 *w/v*), whereas PHA was extracted using a 10 mM NaCl pH 2.4 solution (1/10 *w*/*v*) under the same conditions. Both the suspensions for PHAS and PHA were centrifuged for 30 min at 10,000× *g* and the supernatant was mixed with an equal volume of a buffer of 20 mM Tris-HCl pH 8.6 containing 1% SDS, 8.3% glycerol and 1% DDT. Fifteen microliters of the extracts of the two protein fractions were heat-treated at 100 °C for 5 min and then separated on 15% SDS-PAGE (Sodium Dodecyl Sulphate-Polyacrylamide Gel Electrophoresis) according to Bollini and Chrispeels [49]. The gels were stained with Coomassie Brillant Blue R-250.

### 2.5. DNA Isolation and Microsatellite Analysis

Total DNA was extracted from the four seedlings for each accession obtained from the germination of the part of the seeds with the embryo used for PHAS and PHA investigation by the NucleoSpin^®^ Plant II kit (Macherey-Nagel, Düren, Germany), according to the manufacturer’s instructions. Andean (MIDAS and Jalo EEP558) and Mesoamerican (BAT and G12873) accessions were used as references for a total of 460 (114 accessions x 4 individuals + 4 references) samples. Both 0.8 (*w*/*v*) agarose gel stained with ethidium bromide (0.001%) and a Nanodrop Bioanalyzer ND1000 (Thermo Scientific, Waltham, MA, USA) were used to determine the integrity and concentration of DNA. Twelve SSRs were selected from previous genetic studies of Italian bean collections [27,40,41,42,43] based on their high values of PIC (Polymorphic Information Content) and dispersed map positions (located in 9 of the 11 linkage groups of common beans) (Appendix A) [50,51,52]. PCR reactions were performed as previously reported [53], using primers labelled with FAM, TAMRA, and JOE (Eurofins Genomics, Ebersberg, Germany). Amplification products were checked by agarose gel electrophoresis (1.5%) and the fragments were separated by capillary electrophoresis using an ABI PRISM 3500 Genetic Analyzer (Applied Biosystems, Waltham, MA, USA). Signal peak height and allele sizes were assessed with Gene Mapper 4.0 software (Applied Biosystems).

### 2.6. Data Analyses

For the 66 landraces, the frequency distribution for PHAS and PHA protein patterns and qualitative data relative to seed morphological traits and growth habits were evaluated using JMP PRO 15 (Trial Version ©SAS Institute Inc., Cary, NC, USA). Seed weight data were converted into categorical classes of seed size based on a classification commonly used for beans [27,30]: (1) small seed size (100-seed weight < 25 g), (2) medium seed size (100-seed weight 25–40 g), and (3) large seed size (100-seed weight > 40 g). The likelihood-ratio chi-square test in JMP PRO 15 was used to test for dependence between PHAS and PHA types, and the PHAS type and frequency distribution of qualitative seed morphological traits and growth habits for the 66 landraces. The strength of association was determined by calculating Cramér’s V [54].

The mean and standard error of the four quantitative seed morphological traits (seed length, seed height, elongation index, and weight of 100 seeds) were calculated for each landrace. The coefficient of variation (i.e., the value of the standard deviation of the mean divided by the mean), expressed as a percentage, was used to analyse the variability of the four quantitative traits between and within landraces. All these statistical analyses were performed using JMP PRO 15.

The screening power of SSR loci was evaluated by investigating the PIC index [55] using Power Marker 3.25 [56]. Genetic diversity per locus and population (landrace) was evaluated using the following parameters: the number of observed alleles per locus (N_a_), the effective number of alleles (N_e_), the number of rare (allele frequency < 0.05) and private alleles, expected (H_e_) and observed heterozygosity (H_o_), and the inbreeding coefficient (*F*), by using GenAlEx6 [57] and Power Marker 3.25 [56]. Allelic richness (AR) per landrace was evaluated by FSTAT v2.9.3.2 [58]. The analysis of molecular variance (AMOVA) between and within landraces was performed using a distance matrix and suppressing within the individual analysis, as defined in GenAlEx6 [57]. The variance components were tested statistically by nonparametric randomization tests using 999 permutations.

Phylogenetic analysis was performed to estimate the overall relationship among genotypes, accessions, and landraces. The distance-based dendrogram with Nei’s genetic distance [59] and the UPGMA algorithm were developed using MEGAX [60]. Three different phylogenetic trees were computed considering three distinct groupings of SSR profiles: (1) the whole dataset (456 distinct genotypes); (2) 114 accessions (by grouping the SSR profiles for each accession); and (3) 66 landraces (by grouping the SSR profiles of each accession belonging to the same landrace). The reliability of the topology of the trees was assessed via bootstrapping over 1000 replicates using PAUP* 4.0 software [61]. Wright’s population pairwise fixation index (*Fst*) [62] was also calculated to investigate the genetic divergence among the studied landraces using R/HierFstat [63] and visualized using a heatmap.

The Mantel test [64] was used to assess the degree of correlation between the genetic and geographic distance matrices, as defined in GenAlEx6 [57], using 999 permutations of accession locations among all locations.

Bayesian clustering of the collection studied was performed using STRUCTURE software [65], as described in Mercati et al. [53]. The most likely number of clusters was estimated using the method of Evanno et al. [66], who proposed an ad hoc statistic, Δ*K*, to prevent an overestimation of subgroup number by STRUCTURE. Samples with membership probabilities ≥ 0.80 were assigned to the corresponding subgroup. The likelihood-ratio chi-square test was used to test for dependence between PHAS type/growth habit and cluster membership of the landraces, and the strength of association was determined by calculating Cramér’s V [54].

Based on the results of model-based cluster analyses of SSR loci and PHAS type, the genetic diversity of common bean landraces was evaluated by classifying the landraces into two and three groups. The individuals of the accessions that were found to be of “mixed origin” and those that did not show correspondence between PHAS type and cluster membership in STRUCTURE were excluded from further analysis. The genetic diversity of each group of landraces was determined by calculating N_a_ and AR in FSTAT [58] as well as the number of private alleles, N_e_, H_e_, and H_o_ in GenAlEx6 [57] and Power Marker 3.25 [56]. To determine the significance of the differences in AR, H_e_ and H_o_ among groups, Kruskal–Wallis (among all groups) and Wilcoxon (between all possible pairs of groups) nonparametric tests were performed using JMP PRO 15. AMOVA was carried out to partition the genetic variation between and within groups of landraces, as described above.

## 3. Results

### 3.1. Seed Traits and Growth Habit

A total of nine seed morphological traits, five qualitative and four quantitative, were used to evaluate 114 accessions belonging to 66 putative different landraces collected from different geographical areas of the Lazio region (Appendix A). The relative frequency of seed shape, seed coat colour and pattern, and “seed morphotypes” detected in the 66 common bean landraces are reported in Appendix A. The most widespread seed shape in the collection was cuboid (39%), followed by an oval (33%), kidney (20%), truncate (5%) and round (3%) (Appendix A). Forty out of 66 landraces (61%) showed plain coat seeds, whereas the remaining 26 (39%) presented a patterned seed coat (Appendix A). According to the IBPGR descriptors [45], eight different seed coat colours were attributed to the seeds with no pattern: black, brown, pale to dark, maroon, yellow to greenish yellow, pale cream to buff, pure white, green to olive, and red. White (21%), maroon (20%), and pale cream to buff (9%) were the most frequent plain seed coat colours (Appendix A). By contrast, only three different seed coat patterns were found: striped (24%), followed by the pattern around the hilum (11%), and bicolour (5%) (Appendix A). Combining the five qualitative seed descriptors, 32 different morphotypes were identified (Appendix A), with their relative frequencies ranging from 1.52 to 12% (Appendix A). Seven of the 32 morphotypes (22%) showed a frequency higher than 4% and represented about 51% of the collected seeds.

The accessions assigned to the same landrace based on the local name and the available information showed mainly the same “seed morphotype”, except for the accessions named “Regina”, and three accessions called by the farmers “Suricchio” in which two distinct morphotypes were described (Appendix A). In detail, VE458_FR and VE459_FR, collected from two different farms located in Paliano (FR), displayed the morphotype “red cuboid” (00(12)13), whereas the other (VE457_RM), collected from San Vito Romano (RM), displayed the morphotype “brown, pale to dark reniform” (00214). Because of the different origins and seed morphotypes, we assigned, for the following approaches, three different accessions to two different landraces, named, respectively, Suricchio_1 (VE457_RM) and Suricchio_2 (VE458_FR and VE459_FR). This distinctiveness was confirmed by the subsequent biochemical and molecular analyses (see below).

High diversity across the collection was found through the four quantitative seed traits used (Appendix A). According to the coefficient of variation (CV), the common bean germplasm from Lazio displayed wide phenotypic variation for the 100-seed weight (37%) and low variation for seed height (15%) (Appendix A). The landraces with large seeds (100-seed weight > 40 g) are predominant in the collection (68%) compared to those with medium (100-seed weight 25–40 g) and small (100-seed weight < 25 g) sizes, which represent, respectively, 24% and 8% of the whole germplasm investigated (Appendix A).

Regarding the plant growth habit of the 66 common bean landraces, 41 exhibited an indeterminate growth habit and 25 a determinate one (Appendix A).

### 3.2. Phaseolin (PHAS) and Phytohemagglutinin (PHA) Variability

The SDS-PAGE showed three different electrophoretic patterns for the PHAS protein fraction (Figure 1A and Appendix A), allowing the attribution of the landraces to the Mesoamerican or Andean pool. Each landrace was homogeneous for this character, showing only one specific PHAS type. The C or T PHAS types in the Andean pool were detected in 79% of the 66 analysed landraces, with a higher prevalence of C type (45%), whereas only 14 landraces (21%) displayed the S PHAS type typical of the Mesoamerican pool (Appendix A).

In contrast, a higher variability for the PHA protein fraction was observed, with 11 different patterns distributed across the 66 landraces investigated (Figure 1B). The patterns observed were classified into four major PHA variant groups according to the nomenclature proposed by Brown et al. [67]. The first group (PHA patterns ranging from 1 to 6) was characterized by the presence of two bands with a higher molecular weight, the first of which is lighter than the second (Figure 1B) and has been named TG2 [67]. The second group (PHA patterns 7 and 8) was characterized by one band with a higher molecular weight, very similar to the PHA variant named SG2 [67]. Patterns 10 and 11 represented a third distinct group characterized by the presence of a main band with a molecular weight slightly lower than patterns 7 and 8 (Figure 1B), similar to the variant called VG2 type [67]. Finally, pattern 9 showed a reduced number of polypeptides in the PHA region (Figure 1B), as described in the Pinto UI 111 variety by Brown et al. [67]. This last variant is known for its low content of PHA fraction [68] and lack of agglutinating activity [69].

In contrast to the PHAS, a considerable level of heterogeneity for the PHA protein fraction was detected among the accessions of the same landrace. In particular, eight out of the 66 landraces (12%) showed two or three distinct PHA patterns (Appendix A), one of which usually predominated and was considered representative of frequency data counting. Overall, the TG2 group (PHA patterns 1–6) was detected in 77% of analysed landraces, with patterns 1 (21%) and 3 and 4 (15%) showing the highest frequencies (Appendix A). The second most represented group was the SG2 variant (PHA patterns 7 and 8) with a relative frequency of 15%, in which pattern 8 (14%) was predominant (Appendix A). The remaining three PHA patterns (9, 10 and 11) were the least common among the 66 landraces (8%), with pattern 9, characterized by a reduced quantity of polypeptides in the PHA region, detected in only three landraces: “Arsolana”, “S. Anna”, and “Bianco Rieti” (Appendix A).

### 3.3. Relationship between PHAS and PHA Patterns, and PHAS Pattern and Morphological Data

The association between the PHAS and PHA patterns was highly significant and nearly complete (χ2 = 187.93; df = 20; *p* < 0.0001; Cramér’s V = 0.94). The landraces with the C and T types showed mainly the PHA patterns of the TG2 groups (patterns 1 to 6), whereas the landraces belonging to the Pinto UI 111, VG2, and SG2 groups were characterized by the S type (Figure 2). Within the SG2 group, the PHA pattern 8 represents an exception to the close association found between PHAS and PHA patterns, showing both T- and S type, with a prevalent profile of the latter (12%).

A strong association was also found between PHAS pattern and growth habit (χ2 = 49.9; df = 2; *p* < 0.0001; Cramér’s V = 0.71). Plants with indeterminate growth habits were exclusively present in landraces with the S type. Moreover, plants with indeterminate growth habits were prevalent in the landraces showing the C type, whereas plants with determinate growth habits were prevalent in those containing the T type (Appendix A).

Finally, seed morphological traits were distributed differently among the three PHAS types (Appendix A), with a positive and generally moderate association found between PHAS pattern and seed colour (χ2 = 48.021; df = 16; *p* < 0.0001; Cramér’s V = 0.60), seed shape (χ2 = 28.673; df = 8; *p* < 0.0004; Cramér’s V = 0.39), and seed weight (χ2 = 54.29; df = 4; *p* < 0.0001; Cramér’s V = 0.63).

### 3.4. Genetic Analysis by SSR Markers

#### 3.4.1. Genetic Diversity and Relationships among Landraces

The main genetic parameters for each SSR are reported in Appendix A. A total of 75 alleles were detected at the 12 SSR loci, ranging from 2 (AZ044945 and X74919) to 14 (AF483876 and X61293) alleles per locus, showing a high value of expected heterozygosity (H_e_) (mean 0.622) and considerable discrimination power, with a PIC value > 0.5 for eight SSR used (67%). The mean observed heterozygosity (H_o_) was 0.014 and the inbreeding coefficient (*F*) ranged from 0.951 to 1 due to the absence of heterozygotes in five out of the twelve loci used (Appendix A).

Thirty-six out of 66 landraces had no heterozygous individuals (H_o_ = 0.000) and 34 of them were genetically uniform (H_e_ = 0.000) (Appendix A). Overall, the collection exhibited a significant number of rare alleles (30) (Appendix A). Thirty accessions belonging to 11 different landraces showed 14 private alleles with the landrace “Purgatorio” holding the highest number, corresponding to three private alleles at three different loci (X04001, X80051, and X61293), followed by the landrace “Arsolana”, with two private alleles at two different loci (AF483902 and AF483867) (Appendix A).

Although most of the SSR markers variability was attributable to differences among landraces (89%), AMOVA analysis showed that a considerable fraction of genetic variation (11%) had also occurred within them (Appendix A). Except for AZ044945, intra-landrace variability was detected in all of the SSRs analysed, with a different efficiency degree (Appendix A), allowing the recognition of several unique profiles associated with the genotypes for each landrace, one of which was usually predominant. Obviously, the highest number of profiles was detected for landraces that included more than one accession, ranging from 3 (“Cannellino Atina”) to 13 (“Cannellino Rosso Piumarola”) (Appendix A), with six landraces showing more than eight different genotypes (Appendix A).

Phylogenetic analysis on both the accessions and genotype levels was performed to estimate the inter- and intravariability among landraces included in the germplasm investigated. Overall, all samples belonging to the same landrace were grouped in the same main branches (Appendix A), suggesting that the genetic differences within landraces could represent correct landrace variability.

Phylogenetic analysis grouped the landraces into two well-supported clusters (bootstrap support value higher than 90%, data not shown) corresponding to Mesoamerican and Andean origins of landraces (Figure 3), and in agreement with the PHAS analysis. Indeed, cluster “M” included the two Mesoamerican reference genotypes (BAT and G12873) and 14 landraces with the PHAS S type, characteristic of the Mesoamerican pool. By contrast, the large cluster “A” comprised the two Andean references (MIDAS and JALO) and 52 landraces, which are characterized by the presence of the PHAS C and T types, both related to the Andean pool. Cluster “A” can also be split into two distinct subclusters (“A1” and “A2”) containing mainly the landraces with the C and T types, respectively, although the separation of these two subclusters was not supported by a high bootstrap value (<50%, data not shown). In particular, 22 out of 28 landraces (79%) included in the subcluster “A1” showed the PHAS C type, whereas the subgroup “A2” counted 24 landraces of which 66% (16 out of 24) had the PHAS T type (Figure 3). In the two subclusters of the main cluster “A”, it is also possible to highlight distinct groupings of landraces showing evident relationships with their seed morphotype, name, or geographical origin, such as the “Cannellino” landrace from the province of Frosinone, or three landraces from the province of Viterbo named “Cera”, “Cerino”, and “Verdolino” (Figure 3). It can be reasonably assumed that some of the landraces may have a common origin, being derived from an ancestral population widely cultivated in the past, and are probably the result of the indirect selection carried out by farmers in a given area.

Considering the large number of landraces investigated, only one putative case of synonymy was found, related to two accessions from the Viterbo province named “Occhietto” and “Giallostoppa” (Figure 3). These accessions showed identical molecular profiles, the same seed morphotype (00313), and PHAS and PHA protein patterns (PHAS type T and PHA pattern 4, respectively) (Appendix A), therefore, although they come from two farms located in different municipalities in the province of Viterbo (“Occhietto” at Montefiascone and “Giallostoppa” at Acquapendente), they likely belong to the same landrace.

On the other hand, cases of homonymy were also detected for some couples or groups of landraces. The largest group is represented by the landraces called “Regina”, which in total are nine: four from the province of Roma, three from the province of Rieti, and two from the province of Latina. These nine landraces, despite having the same name or a name that includes the term “Regina”, showed different SSR profiles (Figure 3). Two other cases of homonymy were found for two pairs of accessions collected from the province of Rieti and Viterbo named “Monichelle” and “Giallo”, respectively, suggesting that they belong to the same landrace. The biochemical and molecular analyses indicated, however, that each of the two accessions with the same name could be assigned to the two different gene pools of Andean and Mesoamerican origin. For this reason, the two accessions with the same name were considered as two distinct landraces: Monichelle_1 and Giallo_1, included in the main cluster “A”, and Monichelle_2 and Giallo_2, included in the main cluster “M” (Figure 3).

A high and significant degree of genetic differentiation was found among the 66 landraces by using *Fst* values (Figure 4). Six main clusters were identified (Figure 4, from “a” to “f”), allowing the separation of the landraces included in the Mesoamerican and Andean groups, in agreement with the phylogenetic analysis. Indeed, although the majority of the comparisons between landraces showed *Fst* values greater than 0.5, Andean genotypes with PHAS S type (in pink) were grouped in two distinct branches (cluster “d”, and a subcluster of cluster “e”), whereas the landraces characterized by PHAS T- (red) and PHAS C type (blue) were split into other different branches. Interestingly, as also reported in the UPGMA tree, one main branch (“a”) was nearly accounted for (88%) by samples with the PHAS C type. In contrast, cluster “b” gathered mainly landraces (71%) showing the PHAS T type.

A nonsignificant correlation between geographic and genetic distance matrices was observed for the 114 accessions assigned to the 66 putative different landraces when the Mantel test was applied (r^2^ = 0.0006; r = 0.025 *p* > 0.05).

#### 3.4.2. Genetic Structure of Landraces

The structure analysis of the *P. vulgaris* collection studied confirmed the major clusters found in the phylogenetic analysis. The optimum pool number was recorded at *K* = 2 and *K* = 3 (Appendix A). At *K* = 2, the Andean and Mesoamerican gene pools were clearly distinct (Figure 5). It is interesting to note that all of the 114 accessions belonging to the 66 landraces showed correspondence between PHAS type and the membership according to model-based clustering analysis based on SSR loci (i.e., Mesoamerican group: cluster “M”/PHAS type S; Andean group: cluster “A”/PHAS type T or C). A total of six accessions belonging to five different landraces (8%) could be considered as putative hybrids between Andean and Mesoamerican gene pools, having the membership probabilities Q < 0.8 for both clusters in all the four genotypes analysed for each accession (Figure 5; Appendix A). The genotypes belonging to these landraces were excluded from the subsequent analysis of genetic diversity. A particular case is represented by one of the four accessions belonging to the landrace “Verdolino” (Verdolino_3, VE191_VT), in which only two of the four analysed genotypes can be considered of “mixed origin” with a value of Q < 80% (Figure 5 and Appendix A), indicating a considerable level of heterogeneity within this accession. These two genotypes were also excluded from the genetic diversity analysis.

At *K* = 3, an additional subdivision within the Andean gene pool was highlighted, grouping the accessions based on the PHAS pattern and growth habit (Figure 5). Interestingly, the Mesoamerican cluster (dark violet) included exclusively accessions with an indeterminate growth habit and the S PHAS type (Figure 5). Moreover, the Andean cluster “A1” (blue) consisted mostly of accessions belonging to different landraces of indeterminate growth habit with the C PHAS type (23 out of 29 landraces), whereas the majority of accessions belonging to the Andean cluster “A2” (orange) were of determinate growth habit (19 out of 23 landraces) and contained the T PHAS type (16 out of 23 landraces) (Figure 5). This led to a strong association between cluster membership and growth habit (χ2 = 31.7; df = 2; *p* < 0.0001; Cramér’s V = 0.70) as well as genetic cluster membership and PHAS pattern (χ2 = 81.9; df = 4; *p* < 0.0001; Cramér’s V = 0.79). A total of 15 accessions belonging to 13 different landraces showed an admixture profile (Figure 5 and Appendix A). Furthermore, an additional 23 accessions belonging to 14 landraces did not show the correspondence between PHAS type and the membership according to model-based clustering analysis. As three accessions belonging to three distinct landraces were classified as both “mixed origin” and “noncorresponding”, 35 accessions belonging to 24 landraces (36%) could be derived from hybridization among the three groups and their genotypes, and together with the two from the heterogeneous accession of “Verdolino” landrace (see above), were excluded from the following investigation.

By classifying landraces into two groups (*K* = 2, Mesoamerican group, cluster “M”/PHAS type S vs. Andean group, cluster “A”/PHAS type T or C), the Andean group of landraces showed higher values of AR, H_e_, and H_o_ than the Mesoamerican group, but the differences were not statistically significant following the Kruskal–Wallis test (Table 1). AMOVA analysis revealed that 50% of the SSR markers diversity could be attributed to differences among groups and the remaining 50% within groups (Table 2). This result could ensue partly from the detected intra-landrace variability, but more importantly, from the presence of gene flow between landraces belonging to the two groups.

The diversity analysis of the three groups isolated at *K* = 3 (Mesoamerican group: cluster “M”/PHAS type S; Andean group “A1”: cluster “A1”/PHAS type C; Andean group “A2”: cluster “A2”/PHAS type T) revealed that the Andean group “A1” had the highest AR value, whereas the Andean group “A2” and the Mesoamerican group M showed the highest H_e_ and H_o_ values, respectively. However, the differences among groups were not significant (Table 1). AMOVA analysis based on three groups showed similar results to those recorded at *K* = 2:50% of SSR diversity was attributed to differences among groups (Table 2).

## 4. Discussion

Crop species germplasm, both in situ and ex situ, represent a useful integrated approach for landrace preservation to be employed in breeding programs. In pursuit of new sources for genetic improvement, plant breeders analyse large numbers of accessions. The management of germplasm must focus on a restricted set of accessions while also maximizing available genetic diversity, thus limiting genetic redundancy. Several common bean landraces are still cultivated in Central and Southern Italy, including the Lazio region, mostly in smallholder-farmer systems and farmer-named cultivars [37], which frequently generate a genetic redundancy in the collection [70]. In the present study, morphological seed traits and biochemical and molecular markers have been combined to characterize a collection of 114 *P. vulgaris* accessions belonging to 66 landraces from the Lazio region. The goal was to understand the patterns of bean genetic diversity in this Italian region, while also analysing the intrapopulation variability of landraces and aiming to safeguard agrobiodiversity.

In Lazio, as in other Italian regions, the local name is a “primary label” of genetic diversity, emphasizing a key distinction among landraces. Indeed, local names often refer to seed shape and colour or are directly linked to the location where landraces are traditionally cultivated, or to seed traits and origin simultaneously. In other cases, they refer to the growing period, to some organoleptic characteristics, heirloom status, or to the local celebration where they are traditionally used. However, landraces designated with the same local names from different locations are distinct for some or all the considered morphological traits and biochemical and molecular markers. In this context, the emblematic case is represented by the landraces labelled as “Regina” = queen, a very common name among the bean landraces that, in many cases, is not linked to historical events but to the fact that beans in the past have represented a fundamental food for the people and therefore was considered the “queen dish” of the poors’ table. As previously proposed [39,42,71], the local name, along with the place of origin, may enable a stratified sampling strategy to retrieve all or a major part of genetic diversity, also including landraces with the same name but from separate areas, bearing distinct genetic profiles (homonymies).

### 4.1. Seed Morphological and Biochemical Traits of Lazio Germplasm

A wide variation in the bean collection of the Lazio region was found for colour, size, and shape of the seed, or growth habit, and most of the investigated landraces may be identified based only on their morphology. The characterization of morphological seed traits enabled the identification of 32 “seed morphotypes” of which 18 had a distinct profile. Common bean landraces with the same seed morphotype were distinguished by biochemical and molecular analyses. The level of variability found in this study for seed coat colour and shape characteristics is comparable to that reported in similar studies conducted for the characterization of the germplasm of *P. vulgaris* grown in Sardinia [39] and in Calabria [42], where 29 and 34 different morphotypes were identified by analysing 73 and 87 landraces, respectively. Interestingly, the top three most prevalent morphotypes in our collection, “maroon cuboid” (00313), “white reniform” (00714), and “white oval” (00712) were also among the most frequent morphotypes found in the landraces from previous studies [39,42]. Moreover, three quantitative seed traits (height, length, and 100-seed weight) seemed to be important for landrace distinctiveness. These results indicated that seed morphology may be a useful tool for identifying landraces, as previously reported [46,72,73].

PHAS analysis indicated that both Andean and Mesoamerican gene pools are present in the germplasm of the Lazio region, with a prevalence of the Andean. Indeed, the C or T types, typical of the Andean gene pool, were found in 79% of the landraces, with a higher prevalence of C type (45%). These findings were consistent with those of other Italian regions, such as Basilicata [74], Abruzzo [75], Sardinia [39], Sicily [43], and Calabria [42]. A high prevalence of the C type was primarily found in the countries surrounding the Mediterranean basin, such as the Iberian Peninsula, Italy, and the Balkan area [2]. Conversely, the T type was the most prevalent in France, Central Europe, and Sweden [2]. The similarity in the PHAS pattern distribution between Italian and Spanish germplasm clearly suggests that the common bean was brought to Italy, including the Lazio region, more likely through Spain rather than directly from America. This is hardly surprising considering that many Italian regions were under Spanish control between the sixteenth and seventeenth centuries.

In agreement with previous findings [43,69], our results confirmed a close relationship between specific PHA patterns and the three PHAS types, although gene families coding for PHAS and PHA are not linked with each other [69]. As discussed previously by Lioi [69], this could be an effect of the development of the two separate gene pools in which specific alleles of different genes may be associated more frequently than predicted by random assortments. The geographical distribution of PHAS types suggests that the progenitors of TG2, SG2, and MG2 PHA pattern groups probably originated separately in South- and Meso-America, respectively, and their domestication and expansion followed the history of the PHAS types. Indeed, *P. vulgaris* is a self-pollinating species with little cross-pollination, which could justify the rare occurrence of alternative PHAS and PHA pattern combinations.

Interestingly, the landraces “Arsolana”, “S. Anna”, and “Bianco Rieti”, characterized by the S PHAS type, showed the PHA-deficient pattern of Pinto UI 111, strongly correlated with the absence of agglutinating activity [69]. To the best of our knowledge, in the Italian bean germplasm, this PHA pattern was found only in two Sicilian landraces, “Nero” and “Carrubara Nera”, with black-coloured seeds [43]. Conversely, the seed colour in the three landraces from the Lazio region is completely white, a feature highly appreciated by Italian consumers who associate this colour with a thin tegument and higher digestibility. Therefore, particular attention should be devoted to these landraces due to their potentially higher nutritional value and high level of consumer satisfaction.

### 4.2. Molecular Characterization of Lazio Germplasm

The evaluation of molecular diversity using 12 SSR markers allowed us to detect in our collection a total of 75 alleles with a mean of 6.3 alleles per locus, a higher value than those discovered in the *P. vulgaris* germplasm from Sardinia [39] and Calabria [42], and not so different from those reported in other Italian collections [27,40,41,44]. Furthermore, the mean of He across the 12 SSR loci was higher (0.622) than those observed in other studies carried out on Italian [27,39,40,41,42] as well as European collections [28,29,30], highlighting the high level of genetic diversity present in the common bean germplasm in the Lazio region.

The occurrence of 30 rare alleles, 14 of which were private, found approximately in 62% of landraces, highlighted the high differentiation of the common bean collection here investigated. A major priority in the ARSIAL seed bank will be given to the landraces possessing rare and private alleles, whose presence is an important feature for collection because it has been shown to be informative for genetic conservation [76].

Thirty-two out of the 66 landraces analysed (49%) displayed genetic variability, showing significant diversity both between the different accessions of the same landrace and within a single accession. The detected intragenetic variability recorded from one to six different loci for the landrace (with a range of two to four different alleles/loci) highlighted the presence of several distinct genotypes for each landrace. These results indicated that, despite autogamy and the intralandrace uniformity in seed shape and colour, most of the landraces examined in this study could be considered mixtures of genetically distinct pure lines. Therefore, the sampling of one or few individuals per landrace would not be enough to describe their whole genetic background. However, the UPGMA trees, considering both the 456 distinct genotypes or the 114 accessions, and the STRUCTURE analysis showed that all the genotypes or accessions belonging to the same landrace were grouped within a similar genetic pool, underlining that the genetic background within landraces could reflect their variability. All of these findings imply that strategies to encourage on-farm conservation of common bean landraces within the area of cultivation would be important to preserve genetic variability. Negri and Tiranti [77] demonstrated that common bean landraces reproduced in situ retained the same level of allelic diversity as the original population, while those reproduced ex situ displayed a lower allelic diversity because of the loss of rare alleles, allele fixation, and increased homozygosity. Therefore, farmers must be encouraged to continue selecting and managing local crops by providing incentives and raising the market value of landraces [77,78].

The detected genetic profiles allowed all of the landraces belonging to our collection to be distinguished, with the only exception of two accessions from the Viterbo province, called “Occhietto” and “Giallostoppa”, which likely represent a case of synonymy. Surprisingly, this distinction was unrelated to the geographical separation among landrace cultivation locations, as no significant correlation between the geographic distances and the genetic differentiation among landraces was found. This lack of correlation may be the consequence of an extensive gene flow resulting from traditional seed exchange practices, such as trading between farmers’ families and neighbours or at local markets. These activities resulted in the preservation of the common bean genetic diversity in the Lazio region, as evidenced by the presence of all three detected clusters in the STRUCTURE analysis in each of the five provinces of the region, and by the AMOVA analysis in the classification of the landraces into two and three groups, which showed that 50% of SSRs diversity could be attributed to differences within groups. Furthermore, no significant correlation between genetic and geographical distances was found in domesticated accessions of American bean germplasm, which could be explained by seed-based gene flow following domestication [21]. Seed-based gene flow across distinct landraces and seed exchange between farmers within and outside the cultivation areas are also likely to have occurred and may still be happening on a large scale in the entire Italian peninsula [27]. Seed exchange by farmers is a key element of traditional agricultural systems worldwide [79] and one of the most significant processes of farmers’ management influencing the genetic structure of landrace populations [30,53,80,81].

Overall, the phylogenetic, population structure, and *Fst* analyses of the 66 landraces from the Lazio region reflect the two independent domestication events that occurred, separating into specific branches, the common bean cultivated forms into Mesoamerican and Andean genotypes. Indeed, the optimum number of groups in our structure analysis was *K* = 2, with 15% of landraces of Mesoamerican origin (all characterized by the presence of PHAS S type) and 77% of Andean origin (with a C or T PHAS pattern), whereas 8% of landraces represent putative hybrids between gene pools. Our results are consistent with the findings of previous studies in that the Italian, and, more broadly, European common bean germplasm originates from both Mesoamerican and Andean gene pools, with the latter being more frequently found [27,37,82]. In contrast, it appears that the two gene pools are distributed differently in other regions of the world. For instance, in Africa as a whole, the frequencies of the two gene pools are roughly equivalent, despite substantial differences between countries [83,84,85]. Such differences have been largely attributed to ecological and economic factors that vary from country to country [83]. In China, a preponderance of the Mesoamerican gene pool has been found [26]; however, this has been primarily ascribed to founder effects [26]. In South America, a particular scenario has evolved in Brazil. While Brazil is nearer to the Andes than to Mesoamerica, the Mesoamerican gene pool is largely predominant [86]. Numerous introductions of Mesoamerican genetic material in periods preceding or following the discovery of the Americas could explain this pattern [32].

The well-supported separation between Mesoamerican and Andean groups at *K =* 2 is attributable to the considerable genetic differentiation between these two gene pools, which could conceal other possible substructures in our collection. Indeed, at *K* = 3, the genotypes belonging to the Andean group showed two subpopulations, with a split based on the PHAS pattern and growth habit. Other Italian collections also showed the presence of heterogeneity in common beans of Andean origin. For instance, a study using 12 SSR polymorphic loci classified 146 Italian common bean landraces into three groups: one consisted primarily of Mesoamerican accessions with S-type PHAS, while the others did not show a clear distribution pattern for Andean samples with T and C PHAS types. The authors attributed these profiles to the adaptation to different environmental conditions [27]. Another work on common bean landraces from the Calabria region discovered one and four clusters of Mesoamerican and Andean origin, respectively, as well as a relatively high proportion of accessions arising through inter- and intraspecific hybridizations [42].

A sequence of consecutive bottlenecks during common bean domestication, initial introduction to Portugal and Spain, and subsequent expansion across the Italian peninsula could explain the nearly full consistency of classifications based on PHAS analysis and model-based grouping using SSRs, as well as the strong association between group membership and growth habit in the germplasm from the Lazio region. The Mesoamerican group of landraces together with the Andean group “A1” were largely made up of accessions with indeterminate growth habits, whereas the Andean group “A2” included most landraces with determinate growth habits. Similarly, a strong association between PHAS pattern and growth habit was already reported by Raggi et al. [27] and Carovi’c -Stanco et al. [29] for Italian and Croatian common bean landraces, respectively.

The comparable level of genetic diversity recorded in the present study for landraces of Andean and Mesoamerican origin was previously reported in Iberian and Croatian landraces [29,87] and, in general, at the European level [24], whereas in the domestication centres, the diversity observed in the Mesoamerican gene pool is higher than in the Andean one [21]. Angioi et al. [24] provided two nonexclusive explanations: (i) further selection in Europe might have lowered the variation of the Mesoamerican germplasm, and/or (ii) diversity of Mesoamerican introductions to Europe was already reduced in comparison with the ancestral Mesoamerican gene pool. Furthermore, the apparent incongruence can be attributed to the fact that the Andean gene pool is characterized by two distinct groups of accessions (determinate/indeterminate), which arose from divergent selection during domestication in the Andes [88].

Structure analysis highlighted a low proportion of hybridization (12%) between Mesoamerican and Andean gene pools in the Lazio landraces analysed. This “admixture” level was similar to that observed in a large common bean collection from Italy [27] but significantly lower than the proportion previously detected at the European level [24,34]. Indeed, by analysing 307 European common bean accessions, Angioi et al. [24] estimated that about 44% of them were hybrids of Mesoamerican and Andean gene pools, although the hybridization process was more prevalent in Central Europe than in Italy and the Iberian Peninsula. Furthermore, Gioia et al. [34] found that 40% of 256 European common bean accessions originated through the hybridization of the two gene pools. Therefore, as observed for Italian common bean germplasm at a large scale [27], in landraces from the Lazio region inter-gene pool hybridization seems to be extremely limited, but differences in the methodologies employed to find hybrids must also be considered when comparing our results to those obtained in Europe. In fact, Angioi et al. [24] combined chloroplast SSRs with two nuclear loci (Pv-shatterproof1 and PHAS types), and Gioia et al. [34] utilized chloroplast and nuclear SSRs in addition to two nuclear loci (Pv-shatterproof1 and PHAS types), whereas the present study relied on nuclear SSRs and PHAS analysis.

## 5. Conclusions

This study gives the first thorough overview of the genetic diversity, structure, and distinctiveness of common bean landraces from the Lazio region. The high genetic diversity discovered and analysed in the common bean collection highlights its potential economic value for identifying adaptive characters to environmental stresses and low-input conditions, which are highly prevalent in marginal areas where bean cultivation in Italy, including the Lazio region, is currently confined. Regardless of a low estimated rate of outcrossing and admixture, common beans from the Lazio region retain an appreciable level of intra-landrace diversity, a factor that must be carefully addressed when planning in situ/on-farm conservation strategies. Moreover, morphological and molecular analyses enabled us to identify some redundancies that are relevant for defining an ex situ core collection. Furthermore, the morphological, biochemical, and molecular data obtained in this study can assist the landrace protection schemes that are currently being developed in Italy and could facilitate the registration of the landraces still cultivated in the Lazio region in the National Catalogue of “conservation varieties” that enables their seeds to be commercialized. Both of these activities can aid the future survival of these landraces in situ. Finally, the genetic and morphological characterization of examined common bean landraces could be relevant for their utilization in breeding activities to face climatic changes as well as to meet the need for novel cultivars for marginal lands and/or organic agriculture.

## Figures and Tables

**Figure 1 plants-12-00744-f001:**
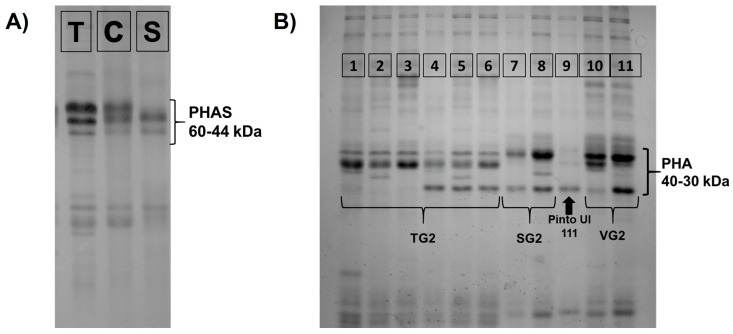
PHAS and PHA protein patterns detected across the 66 common bean landraces in the Lazio region. (**A**) SDS–PAGE of PHAS protein fraction showing the T, C, and S types. (**B**) SDS–PAGE of the PHA protein fraction showing 11 different patterns, which can be divided into four major variant groups according to the nomenclature proposed by Brown et al. [67]: TG2, SG2, PintoUI111, and VG2.

**Figure 2 plants-12-00744-f002:**
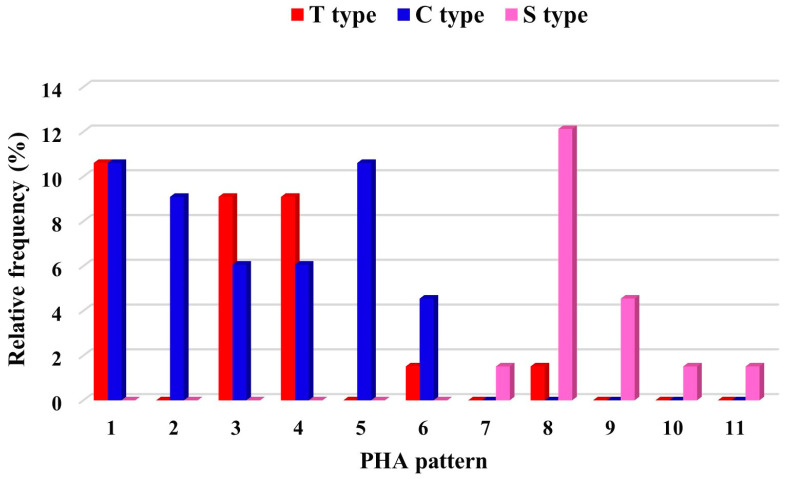
Frequency distribution of PHA patterns in relationship with PHAS types in the landrace collection from the Lazio region.

**Figure 3 plants-12-00744-f003:**
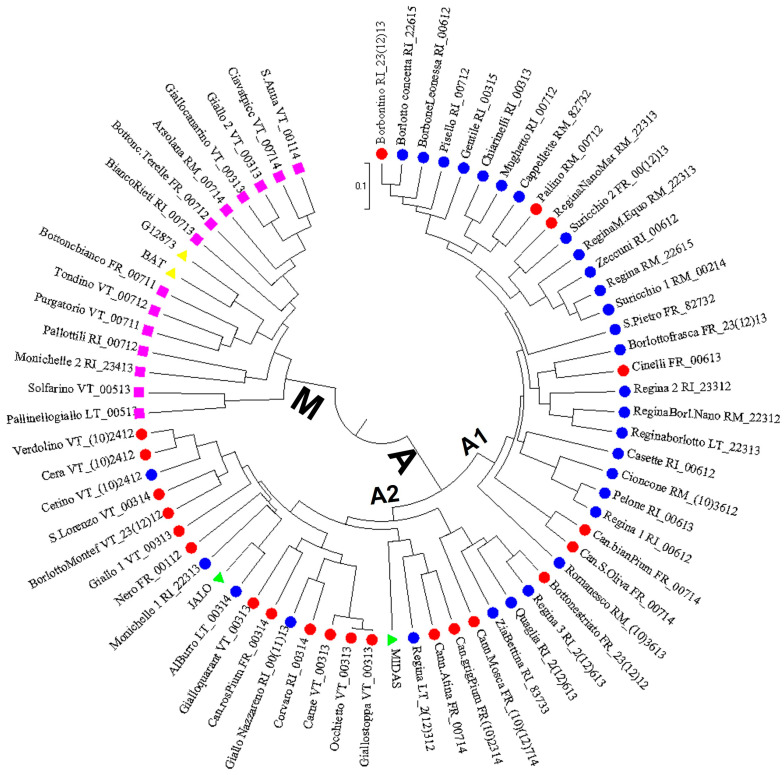
Dendrogram of the genetic relationships among the 66 *P. vulgaris* landraces from the Lazio region generated with Nei’s coefficient [59] and UPGMA cluster analysis. In the phylogenetic tree, the blue and red circles and the pink square indicate the landraces with C, T, and S PHAS types, respectively. The genotypes BAT and G12873 of Mesoamerica origin (yellow triangles) and MIDAS and JALO of Andean origin (green triangles) were included as references. The numerical code for the seed morphotype is also reported for each landrace.

**Figure 4 plants-12-00744-f004:**
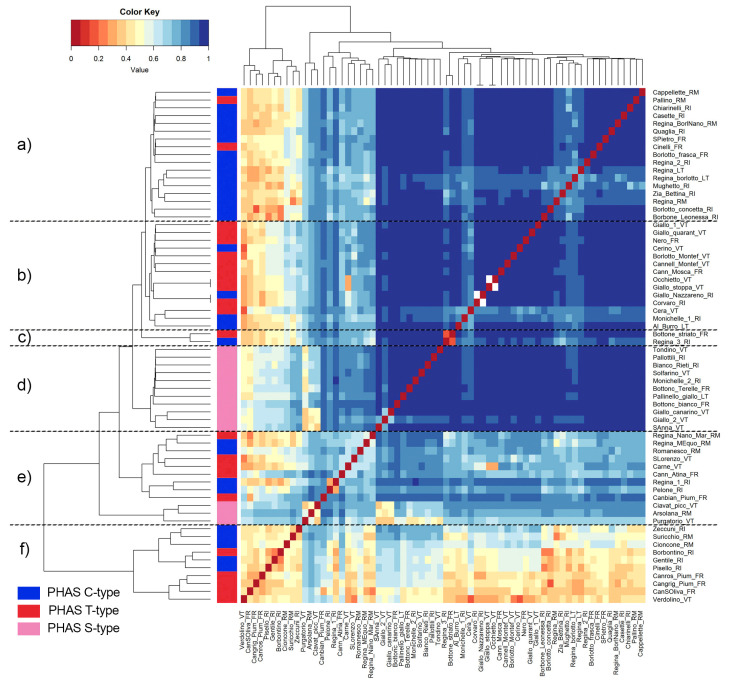
Pairwise *Fst* heatmap and dendrogram based on *Fst* values among the 66 landraces of *P. vulgaris* collected in the Lazio region. The colour key represents the *Fst* matrix considering different discrete *Fst* bins, from low (red) to high (blue) genetic differentiation. Phylogenetic trees were developed using the UPGMA method. Different letters (from “(**a**)” to “(**f**)”) indicate the main six clusters identified.

**Figure 5 plants-12-00744-f005:**
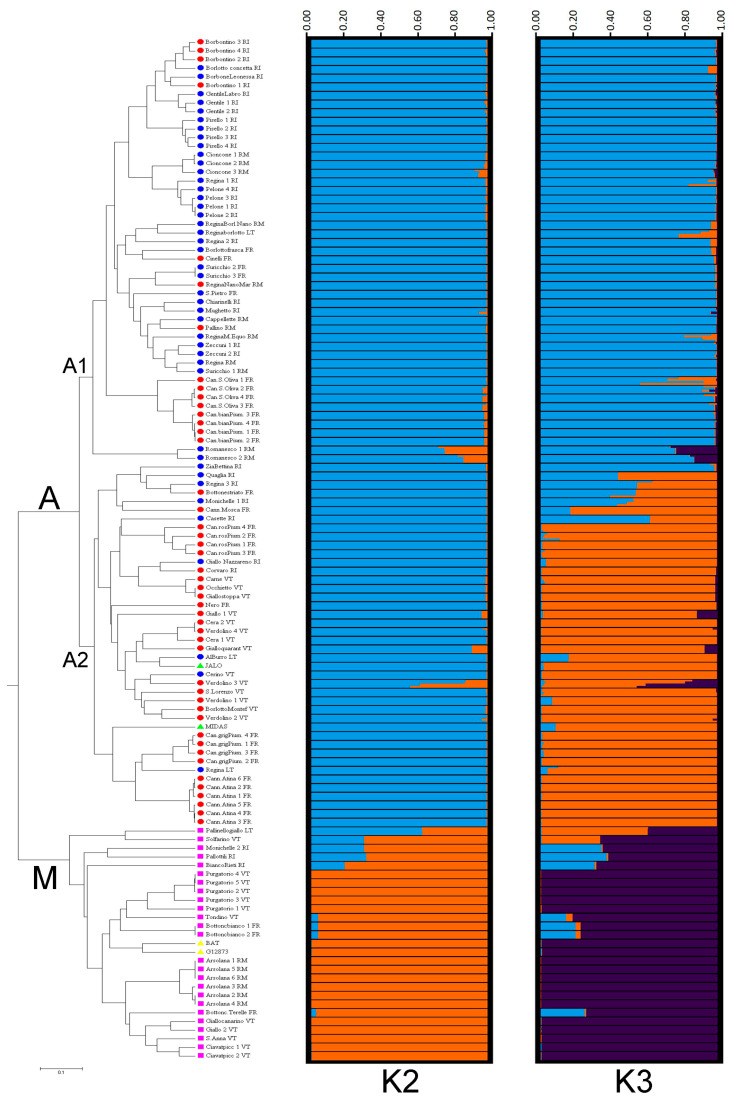
UPGMA tree based on Nei’s coefficient [59] among the 114 accessions belonging to the 66 *P. vulgaris* landraces from the Lazio region. Structure bar plots of average proportions of membership for K = 2 (in blue and orange) and K = 3 (in blue, orange, and dark violet) clusters are given for the four genotypes of each of the 114 accessions studied. In the phylogenetic tree, the blue and red circles and the pink square indicate the landraces with C, T, and S PHAS types, respectively. The genotypes BAT and G12873 of Mesoamerican origin (yellow triangles) and MIDAS and JALO of Andean origin (green triangles) were included as references.

**Table 1 plants-12-00744-t001:** Genetic diversity of landraces classified into two and three groups according to model cluster analysis and PHAS type. N: number of individuals for each group; N land: number of landraces for each group; N Ind: number of landraces with indeterminate growth habit; N Det: number of landraces with determinate growth habit; Na: number of alleles per locus; Ne: number of effective alleles; Npa: number of private alleles; AR: allelic richness; H_o_: observed heterozygosity; H_e_: expected heterozygosity.

Cluster/Phaseolin Type	N	N Lan	N Ind	N Det	Na	Ne	Npa	AR	H_o_	H_e_
M/PHAS type S (Mesoamerican)	84	10	10	0	3.500	2.212	10	3.500	0.010	0.382
A/PHAS type T or C (Andean)	346	51	26	25	4.830	2.593	38	4.660	0.016	0.466
*p*-value Test Kruskal–Wallis								0.199	0.266	0.174
M/PHAS type S (Mesoamerican)	68	7	7	0	3.000	1.969	9	3.000	0.028	0.337
A1/PHAS type C (Andean)	128	20	18	2	3.833	2.087	13	3.802	0.018	0.382
A2/PHAS type T (Andean)	118	15	1	14	3.667	2.226	7	3.625	0.023	0.422
Wilcoxon Test *p*-value										
M/A1								0.320	0.220	0.622
M/A2								0.398	0.530	0.283
A1/A2								0.931	0.490	0.452

**Table 2 plants-12-00744-t002:** Analysis of molecular variance for the partitioning of SSR markers diversity of landraces classified into two (A) as well as three groups (B) according to model-based cluster analysis and PHAS type. P(*Φ*)–*Φ*-statistical probability level after 999 permutations.

Analysis	Source	df	SS	MS	Est. Var.	%	*Φ*-Statistic	P(*Φ*)
(A)	Among groups	1	1518.301	1518.301	11.149	50	0.501	<0.001
	Within groups	428	4756.799	11.114	11.114	50		
	Total	429	6275.100		22.263	100		
(B)	Among groups	2	1861.247	930.623	9.091	50	0.503	<0.001
	Within groups	311	2796.690	8.993	8.993	50		
	Total	313	4657.936		18.084	100		

## Data Availability

The data contained within the present article and in its Appendix A are freely available upon request to the corresponding author.

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
