# Peer review of "Genetic Diversity and Population Structure of Common Bean (Phaseolus vulgaris L.) Landraces in the Lazio Region of Italy"

_plants, 2023, doi:10.3390/plants12040744_

Round 1
Reviewer 1 Report
The manuscript entitled Genetic diversity and population structure of common bean (Phaseoulus vulgaris L.) landraces from the Lazio region (Italy) presents new insights about molecular, morphological and biochemical results, which can be further be used for in situ/ex situ protection. I found the manuscript well written and methodology quite analytically presented. I only spotted that an "o" is missing from protein in line 186. Therefore I have no other suggestions to improve the manuscript.
Author Response
Response to Reviewer 1 Comments
Point 1: The manuscript entitled Genetic diversity and population structure of common bean (Phaseoulus vulgaris L.) landraces from the Lazio region (Italy) presents new insights about molecular, morphological and biochemical results, which can be further be used for in situ/ex situ protection. I found the manuscript well written and methodology quite analytically presented. I only spotted that an "o" is missing from protein in line 186. Therefore I have no other suggestions to improve the manuscript.
Response 1: Firstly, I would like to thank Reviewer 1 for the positive assessment of the work and following his/her suggestion we have corrected the typo by replacing “prtein” with “protein” (Line 205 of the present revised version).

Reviewer 2 Report
[MAJOR] Please have your manuscript proof-read by a native-English speaker with experience in the discipline.
56: Support with statistics aggregates: FAOSTAT? EuroStat?
89-91 What bigger-picture conclusion does this association suggest?
[MAJOR] Introduction jumps with topics in each paragraph; to make it less repetitive (and shorter!), put similar things together. This will help with the flow of thought in this part as well.
163-166 Please provide source (collection?) of the S American accessions. If previously published by your group or by others, in this or related aspect(s), please cite.
175 100 seed weight: Done in how many replications?
186 "prtein"
246-247 [MAJOR] Mantel test can be used to assess correlation between any two (or more) matrices of distances. What is the strength of such correlation between the phenotypic and the genotypic datasets? What conclusions arise from such analysis?
228-235 and 260-266 Another example of disjointed writing. Another software capable of calculating these indices, their significance using several approaches (and many more!), is Spagedi (Hardy, Vekemans 2002).
284 "ranging from 1.52 to 12% (Figure S2C)" is unclear: do you mean "ranging in frequency of distribution"?
299,300 What do the 37% and 15% mean? How to read these values?
304-305 How was the growth habit determined? This information is lacking in M&M.
388-390 It is unclear, whether 3 (or 2) PA total are meant, or 9 (or 4), as products of PA x loci. Suppl.Table makes this clear, but the data description here is ambiguous.
Figure3 [MAJOR] In the rooted tree presented, what was used as outgroup, to properly root the tree? No such information is presented anywhere.
Figure4 [MAJOR] In the Figure caption, how was the tree developed? How was the clades separation established? What does the color key denote? On the tree, it would help to mark the M/A(-1/-2) clades, for symmetry with other figures. Keeping the PHAS color palette consistent with the other Figures will help.
456-465 [MAJOR] Authors are presenting the PHAS type in relation to Fst. It is unclear why this was chosen as a reliable index to be associated - instead of PHA or others. Arguably, all assessed parameters by Fst (or Nei) correlations analyzed independently should be a basis for such choice.
[MAJOR] Fig3 and Fig5A present the same trees. This is redundant, duplicate presentation. The comment regarding rooting remains valid here.
Table1 [MAJOR] Results of this table are definitely largely ignored, and could be much better presented in the Results part. What is the information for breeding/conservation (as discussed in Introduction) based on Ne, Npa, AR, Ho/He?
[MAJOR] Why does the number of accessions and all related information change between 2 and 3 groups, for M/PHAS type S? From the description, it seemed only the A/PHAS was subdivided.
[MAJOR] One additional suggested analysis would be, combining all assessed parameters and analyze such dataframe using DAPC. This will allow grouping based on all data, and identification of 2 or 3 (if 3-dimensional DAPC) main components with their underlying contributions that explain the dataset's variance.
[MAJOR] Please refrain from presenting the results again in Discussion. This counts as redundant data presentation, but more importantly does not add to the Discussion at all. Major "big-picture" data trends may be recalled, to then be put in context with relevant other studies.
613-615 [MAJOR] This information should be presented in Introduction, as explains practically fully the low Ho observed.
673-685 [MAJOR] Another ignored explanation would be related to the selected SSRs. Are the loci in which they reside under any selection pressures?
Author Response
Response to Reviewer 2 Comments
Point 1: [MAJOR] Please have your manuscript proof-read by a native-English speaker with experience in the discipline.
Response 1: According to the suggestion of the Reviewer 2, the old version of the manuscript was checked for language by a native English-speaker. She was thanked in the Acknowledgments section.
Point 2: [MAJOR] Introduction jumps with topics in each paragraph; to make it less repetitive (and shorter!), put similar things together. This will help with the flow of thought in this part as well.
Response 2: Following the suggestion from the Reviewer 2, we have deleted or modified some parts of the Introduction (see for instance Lines 63-67, 86-101, 112-115 and 135-138 of the present revised version). We believe that after these changes, the flow of information will be clearer and more consequential in order to meet the Reviewer's request.
Point 3: 56: Support with statistics aggregates: FAOSTAT? EuroStat?
Response 3: Done (lines 61 and 63 of the present revised version, ISTAT).
Point 4: 89-91 What bigger-picture conclusion does this association suggest?
Response 4: This sentence was deleted in the new version of the Introduction (see Lines 94-96 of the present revised version).
Point 5: 163-166 Please provide source (collection?) of the S American accessions. If previously published by your group or by others, in this or related aspect(s), please cite.
Response 5: Done (Lines 176-177 of the present revised version).
Point 6: 175 100 seed weight: Done in how many replications?
Response 6: The weight of 100 seeds (g) was determined using two random samples of seeds for each accession. To take this into account, we have inserted a sentence in the subheading “ 2.2. Morpho-Phenotypic Seed Analysis” (Lines 194-195 of the present revised version).
Point 7: 186 "prtein"
Response 7: We have corrected the typo by replacing “prtein” with “protein” (Line 205 of the present revised version).
Point 8: 246-247 [MAJOR] Mantel test can be used to assess correlation between any two (or more) matrices of distances. What is the strength of such correlation between the phenotypic and the genotypic datasets? What conclusions arise from such analysis?
Response 8: We thank the Reviewer for the comment. We are aware that the Mantel test can be used to asses correlation between different matrices of distances. As stated in MM, in this study we used the Mantel test to evaluate the degree of correlation between the genetic and geographic distances and the consequences of the results obtained were largely discussed (see Lines 701-719 of the present revised version). Taking into account the limited number of morphological traits considered, we believe that the analysis of the correlation between genetic and morphological distances cannot provide any more useful information than that obtained from the extensive illustration and interpretation of results reported in this study.
Point 9: 228-235 and 260-266 Another example of disjointed writing. Another software capable of calculating these indices, their significance using several approaches (and many more!), is Spagedi (Hardy, Vekemans 2002).
Response 9: We thank the Reviewer for above comment, even if we were unable to understand its meaning and implications in full. In the two indicated paragraphs (Lines 228-235 and Lines 260-266 of the old version of the manuscript) are reported the parameters used to analyze the genetic diversity in two different groups of data. Although most of these parameters are the same, in the first paragraph we consider those used for the analysis of the genetic diversity per locus and landrace, while in second those used for the analysis of genetic diversity of two or three groups of landraces, according to the results of model-based cluster analysis of SSRs (STRUCTURE) and PHAS type. Furthermore, in these two paragraphs are listed the software/packages (commonly used in many studies and pubblished papers) employed to evaluate the genetic diversity parameters in the plant material studied. Spagedi provides the same parameters, therefore it would be redundant re-run the analysis with this software. In addition to Spagedi, there are other options (e.g. using R packages), but all run with the same algorithms therefore new analysis would provide the same results.
Point 10: 284 "ranging from 1.52 to 12% (Figure S2C)" is unclear: do you mean "ranging in frequency of distribution"?
Response 10: According to the reviewer suggestion, the sentence in question was modified to make it clearer as follow: “Combining the five qualitative seed descriptors, 32 different morphotypes were identified (Table S3 and Figure S3), with their relative frequencies ranging from 1.52 to 12% (Figure S2C)” (Lines 307-308 in the present revised version of the manuscript).
Point 11: 299,300 What do the 37% and 15% mean? How to read these values?
Response 11: These values refer to the coefficient of variation (CV) of the two traits with the highest (100-seed weight) and lowest (seed height) phenotypic variation. As indicated in the subheading “ 2.6. Data Analyses”, the coefficient of variation (i.e., the value of the standard deviation of the mean divided by the mean), is usually expressed as a percentage.
Point 12: 304-305 How was the growth habit determined? This information is lacking in M&M.
Response 12: Following the suggestion from the Reviewer 2, we have included a short paragraph in the MM section describing how the growth habit of the 114 accessions was determined (Lines 197-201 in the revised version of the manuscript).
Point 13: 388-390 It is unclear, whether 3 (or 2) PA total are meant, or 9 (or 4), as products of PA x loci. Suppl.Table makes this clear, but the data description here is ambiguous.
Response 13: We thank the Reviewer for above comment, even if we were unable to understand its meaning. This is the indicated sentence: Thirty accessions belonging to 11 different landraces showed 14 private alleles with the landrace “Purgatorio” holding the greater number, corresponding to three private alleles at three different loci (X04001, X80051 and X61293), followed by the landrace “Arsolana” with two private alleles at two different loci (AF483902 and AF483867) (Table S6 and Table S7). In this sentence is reported that 11 landraces contained 14 private alleles, with the landrace Purgatorio (three private alleles) and Arsolana (two private alleles) showing the higher number, as also clearly reported in Table S6 and Table S7.
Point 14: Figure3 [MAJOR] In the rooted tree presented, what was used as outgroup, to properly root the tree? No such information is presented anywhere.
Response 14: To develop a rooted tree there are several methods and, in more of them, it is not mandatory include an outgroup. In our analysis, we analyzed genetic relationships between the considered landraces, including BAT and G12873 of Mesoamerica origin, and MIDAS and JALO of Andean, as reference genotypes. All these information are already reported in the text and in all captions.
Point 15: Figure 4 [MAJOR] In the Figure caption, how was the tree developed? How was the clades separation established? What does the color key denote? On the tree, it would help to mark the M/A(-1/-2) clades, for symmetry with other figures. Keeping the PHAS color palette consistent with the other Figures will help.
Response 15: We thank the Reviewer 2 for the above comment. As reported in MM Figure 4 shows the Wright’s population pairwise fixation index (Fst) values. The pairs between landraces have been visualized by an heatmap showing the clustering among samples. This clustering has been developed on the basis of Fst values (reported in the color key); it is in agreement to the genetic distance reported on the phylogenetic analysis, but didn’t overlap. Therefore, we cannot report the same clades here. This is a common visualization used, and reported in several pubblished papers.
Following the suggestion from the Reviewer 2, in the present revised version of the manuscript in the Figures 2, 3, 4 and 5 we used the same colors for the different PHAS types (blue, red and pink for C, T and S PHAS types, respectively).
Point 16: 456-465 [MAJOR] Authors are presenting the PHAS type in relation to Fst. It is unclear why this was chosen as a reliable index to be associated - instead of PHA or others. Arguably, all assessed parameters by Fst (or Nei) correlations analyzed independently should be a basis for such choice.
Response 16: We thank the Reviewer 2 for the above comment. We visualized the PHAS profiles for each samples to underline that the Fst and phylogenetic investigations are in agreement, to underline again the strength of the data presented.
Point 17: [MAJOR] Fig3 and Fig5A present the same trees. This is redundant, duplicate presentation. The comment regarding rooting remains valid here.
Response 17: We thank the Reviewer 2 for the above comment. Hovewer, the two figures do not show the same phylogenetic trees. Figure 3 shows the genetic relationships between the 66 landraces: the phylogenetic tree was obtained by grouping the SSR profiles of each accession belonging to the same landrace. Figure 5 shows the genetic relationships between the 114 accessions: the phylogenetic tree was obtained by grouping the SSR profiles of the four individuals analysed for each accession. Since Structure bar plots for K=2 and K=3 are given for the four genotypes of each of the 114 accessions, we consider appropriate to leave the tree for the 114 accessions in Figure 5, in order to have a precise correspondence between each accession and the corresponding bar plot.
As far as the comment on rooting of the tree is concerned, we hope we were able to reply to it convincingly in our previous response regarding the point 14.
Point 18: Table1 [MAJOR] Results of this table are definitely largely ignored, and could be much better presented in the Results part. What is the information for breeding/conservation (as discussed in Introduction) based on Ne, Npa, AR, Ho/He?
Response 18: Concerning Table 1, in the Results section we considered the data of the most important genetic parameters such as AR, Ho and He, for which we determined the statistical significance of the differences between groups. We calculated and reported in the Table also Na and Ne for the two or the three groups only for comparison with AR that is the measure of the number of allelles per locus independent of sample size. The results in the Table 1 that showed a comparable level of genetic diversity for the landraces of Andean and Mesoamerican origin in our collection were then given due consideration in the "Discussion" section (see Lines 764-774 in the present revised version).
Point 19: [MAJOR] Why does the number of accessions and all related information change between 2 and 3 groups, for M/PHAS type S? From the description, it seemed only the A/PHAS was subdivided.
Response 19: We thank the Reviewer 2 for the above comment. Firstly, we would like to highlight that in the old version of Table 1 there was an inversion of the number of landraces with indeterminate (1) and determinate (14) growth habit for the A2 group, which has been corrected in the current version of the Table. We apologise for this error, which probably caused confusion in the reviewer's interpretation of the results.
The number of accessions and their genetic parameters change from two to three groups because the structure analysis for K=3 revealed the presence of a greater number of accessions that could be considered as putative hybrids between the Andean and Mesoamerican gene pools having the membreship probability Q<0.8 for the three clusters in all the four genotypes analysed for each accession as shown in Figure 5 and clearly reported in Table S10. In fact, in addition to the six accessions belonging to five landraces identified for K=2, the structure analysis at K=3 showed that another four accessions belonging to three distinct landraces could be considered as putative hybrids between the two gene pools. We consider the sentences in lines 524-529 of the present revised version of the manuscript to be sufficiently clear to explain the belonging of the different landraces to the three groups M/PHAS S, A1/PHAS C, A2/PHAST in Table 1. However, if the reviewer deems it appropriate we could add the following sentence:
Overall at K=3, the classification of 66 landraces from the Lazio region would the following: 7 landraces (10.61%) belonged to Mesoamerican group, 20 (30.30%) to Andean group “A1”, 15 (22.73%) to Andean group “A2”, 6 landraces (9.1%) were putative hybrids between Mesoamerican group and Andean group “A1”, 2 (3.03%) were hybrids between Mesoamerican and Andean group “A2”, while 16 landraces (24.2%), of which five of “admixture” and 11 “non-corresponding”, were putative hybrids between Andean groups “A1” and “A2”.
Point 20: [MAJOR] One additional suggested analysis would be, combining all assessed parameters and analyze such dataframe using DAPC. This will allow grouping based on all data, and identification of 2 or 3 (if 3-dimensional DAPC) main components with their underlying contributions that explain the dataset's variance.
Response 20: We already run the DAPC approach using R/adegenet package during the paper preparation, but the results didn’t add any useful information. In addition, since our goal was to underline the structure of landraces without a prior information, to correlate these results to the other approaches (e.g. phylogenetic analysis), the STRUCTURE approach is the best one.
Point 21: [MAJOR] Please refrain from presenting the results again in Discussion. This counts as redundant data presentation, but more importantly does not add to the Discussion at all. Major "big-picture" data trends may be recalled, to then be put in context with relevant other studies.
Response 21: Following the suggestion from the Reviewer 2, we deleted some of repetitions of result data that really appear reduntant in the “Discussion” Section (see Lines 620-621, 631-633, 663-667, 676-678 and 688-693 in the present revised version).
Point 22: 613-615 [MAJOR] This information should be presented in Introduction, as explains practically fully the low Ho observed.
Response 22: After the revision of the Discussion section, the paragraph containing this information was deleted as it was considered reduntant and not relevant for the discussion of the obtained results.
Point 23: 673-685 [MAJOR] Another ignored explanation would be related to the selected SSRs. Are the loci in which they reside under any selection pressures?
Response 23: We thank the Reviewer for the comment. Unfortunately, we did not run any specific attempt to verify if the SSR loci used in this study are putatively under any selection pressure, because it was outside the main scope of the present study, which was to investigate the genetic variation, structure and distinctiveness of the analysed landraces. Hovewer, according to the Reviewer, this would be a very interesting and important aspect to be considered in further work.

Reviewer 3 Report
The paper is presents a robust genetic, morphological, and biochemical survey of phaseolus landraces cultivated in the Italian Lazio region. The study records distinctiveness between landraces, including their origins, frequency of hybridization, and within race diversity, helping to recognise the importance of traditional agricultural practise that still occurs in this region.
The abstract summarises the study well, although the aims of the study are not clearly stated at the start but are instead described alongside key results. Beyond describing the diversity of local phaseolus landraces in this region, i assume there is a conservation aim? This could be stated more clearly in the abstract. The introduction provides comprehensive and well researched background to the study system. The language is sometimes a bit awkward but minor editing would fix this. I make some specific suggestions later. Study aims are clearly stated at the end of the introduction. The study is mainly incremental, by better characterising the landraces from a particular region but the value of this work is well justified. Methods are mostly well described, although a little more detail about morphological and protein measures are needed as described in specific comments. The results go into appropriate detail and highlight informative patterns of diversity among the landraces. The discussion starts with some interesting consideration of the value of the cultural context of landraces. The discussion compares the results of this study to similar studies of phaseolus diversity mainly in other Italian regions. It could broaden the interest of the paper to compare to studies of other regions outside Europe also. The discussion about within-landrace variability leads to interesting interpretations about the importance of in-situ landrace conservation. The conclusions show very well how the results of this study could influence conservation plans for phaseolus landraces in the region.
Specific comments
L20-21 Consider to drop "complex"? No reason why these changes have to be complex.
L37-38 Rather than "limited information", I would suggest "incomplete information" as a lot can be learned from a single accession per landrace.
L106 Replace "old continent" with "Europe" to avoid confusion.
L110 Replace "since the past years" with "for many years"
L185-192 The protein analysis section does not describe how proteins were quantified after gel staining. Was there correction for input mass?
L186 Typo: Replace "prtein" with "protein"
L195 The previous section stated that seeds not seedlings were analysed. Which was it?
L201-203 What aspects of PIC and map location were favourable for SSR selection? Specify that you mean genomic map location here.
L220 Growth habit was not listed as one of the measured traits in the earlier section. It would be helpful to reference the supplementary table listing all of the measured traits in addition to the provided references.
L278-279 No need to add colour number codes here as they are confusing with the other counts and percentages that you are presenting.
L304-305 See my earlier comment about growth habit.
L373-376 This repetition from the methods could be removed.
Table 1. Why does the count of determinate growth landraces go from 25 to 3 when splitting from two clusters to three?
L627-634 I think these genetic diversity comparisons should be made cautiously as the different sets of SSRs or other markers used in each study might contribute to differences too.
L639-644 Again, I would be cautious about overstating the importance of rare alleles as 12 SSRs represents a tiny snapshot of overall genetic diversity.
Author Response
Response to Reviewer 3 Comments
Point 1: The abstract summarises the study well, although the aims of the study are not clearly stated at the start but are instead described alongside key results. Beyond describing the diversity of local phaseolus landraces in this region, I assume there is a conservation aim? This could be stated more clearly in the abstract.
Response 1: Following the suggestion from the Reviewer 3, we inserted a sentence in the abstract (lines 25-26 of the present revised version) in order to give due consideration to the conservation aim of the studied landraces.
Point 2: The discussion compares the results of this study to similar studies of phaseolus diversity mainly in other Italian regions. It could broaden the interest of the paper to compare to studies of other regions outside Europe also.
Response 2: Following the suggestion from the Reviewer 3, we tried, where possible, to compare the results obtained in the present study with those of worldwide collections outside Europe (see for instance lines 711-714, lines 716-719 and lines 729-739 of the present revised version).
Point 3: L20-21 Consider to drop "complex"? No reason why these changes have to be complex.
Response 3: Done (Line 21 of the present revised version).
Point 4: L37-38 Rather than "limited information", I would suggest "incomplete information" as a lot can be learned from a single accession per landrace.
Response 4: Done (Line 40 of the present revised version).
Point 5: L106 Replace "old continent" with "Europe" to avoid confusion.
Response 5: Done (Line 111 of the present revised version).
Point 6: L110 Replace "since the past years" with "for many years".
Response 6: Done (Line 118 of the present revised version).
Point 7: L185-192 The protein analysis section does not describe how proteins were quantified after gel staining. Was there correction for input mass?
Response 7: We thank the Reviewer for the comment. The analysis performed in SDS PAGE is not a quantitative but a qualitative analysis that allows the detection of the different patterns of the two protein fractions (PHAS and PHA) based on two different extraction methods. Therefore, to better clarify this concept, we have inserted a paragraph in the the subheading “2.4 Phaseolin (PHAS) and Phytohemagglutini (PHA) Analysis” of the MM section, where the two extraction methods for obtaining the two protein fractions are described in detail (lines 208-213 of the present revised version). Hopefully, this will make the methodology for detecting the different PHAS and PHA patterns clearer.
Point 8: L186 Typo: Replace "prtein" with "protein"
Response 8: Done (Line 205 of the present revised version).
Point 9: L195 The previous section stated that seeds not seedlings were analysed. Which was it?
Response 9: We apologise for the lack of clarity of the sentence reported in L195 and for this we thank the Reviewer for the above comment. As stated in the previous section each of the four seeds for each accession was cut into two parts and the half without embryo was used for PHAS and PHA extraction, while the part with the embryo was germinated to obtain the seedlings that were used to extract the DNA. That is why we modified the sentence as follows: “Total DNA was extracted from the four seedlings for each accession obtained from the germination of the part of the seeds with embryo used for PHAS and PHA investigation by”….. (Lines 219-220 of the present revised version).
Point 10: L201-203 What aspects of PIC and map location were favourable for SSR selection? Specify that you mean genomic map location here.
Response 10: The SSR primer pairs were selected from previous studies based on their high Polymorphic Information Content and dispersed map positions because they are located in 9 of the 11 linkage groups of common beans. According to the reviewer suggestion, the sentence in question was modified to make it clearer (Lines 228-229 in the revised version of the manuscript).
Point 11: L220 Growth habit was not listed as one of the measured traits in the earlier section. It would be helpful to reference the supplementary table listing all of the measured traits in addition to the provided references.
Response 11: Following the suggestion from the Reviewer 3, we have included a short paragraph in the MM section describing how the growth habit of the 114 accessions was determined (Lines 197-200 in the revised version of the manuscript).
Point 12: L278-279 No need to add colour number codes here as they are confusing with the other counts and percentages that you are presenting.
Response 12: Done (Lines 302-303 in the revised version of the manuscript).
Point 13: L304-305 See my earlier comment about growth habit.
Response 13: We thank the Reviewer for the comment, to which we hope we were able to reply convincingly in our previous response concerning the point 11.
Point 14: L373-376 This repetition from the methods could be removed.
Response 14: Following the suggestion from the Reviewer 3, the paragraph between the lines 389-392 in the old version of the manuscript has been removed.
Point 15: Table 1. Why does the count of determinate growth landraces go from 25 to 3 when splitting from two clusters to three?
Response 15: We thank the Reviewer for the comment. Actually, in the old version of Table 1 there was an inversion of the number of landraces with indeterminate (1) and determinate (14) growth habit for the A2 group, which has been corrected in the current version of the Table. We apologise for this error, which probably caused confusion in the reviewer's interpretation of the results.
Point 16: L627-634 I think these genetic diversity comparisons should be made cautiously as the different sets of SSRs or other markers used in each study might contribute to differences too.
Response 16: We thank the Reviewer for the comment. We are aware that comparing estimates of genetic diversity parameters, such as the mean number of alleles per locus and the expected heterozygosity (He), between different studies may often not be appropriate due to the different sets and number of microsatellites used. However, in our case we compared these parameters with studies that shared many of the SSRs used in the present study, as indicated in the MM. We therefore consider these comparisons to be quite indicative of the genetic diversity between our collection and those of the cited studies.
Point 17: L639-644 Again, I would be cautious about overstating the importance of rare alleles as 12 SSRs represents a tiny snapshot of overall genetic diversity.
Response 17: Altough the 12 SSRs “represent a tiny snapshot of overall genetic diversity”, they were very informative and useful to distinguish among common bean landraces from Lazio region, as also evidenced by the high number of rare alleles (30 on the total 75 alles) detected across the 12 SSR loci. As discussed in the manuscript, we wanted to emphasise the importance of the presence of rare alleles only for the conservation of the landraces under study.

Round 2
Reviewer 2 Report
Dear Authors,
Thank you for your efforts on improving this submission. At this stage, I still have several suggestions to try make it an even stronger story. Many of these are minor in character, but several major [MAJOR] corrections are needed, in my opinion.
56,299,316,338,377,441,487,726 while -> whereas
129 Please unify the use of serial comma (Oxford comma) throughout.
146 points of view -> traits(?)
149-150 EC PDO abbreviation unknown.
163 whole -> entire(?)
199-200 To keep things more legible, please refrain from juggling the terms or using synonyms. Is "indeterminate" type II? How does that compare to bushy / climbing used later on (384-385)? This paper is complicated enough that adding extra layers only confounds the possible confusion.
209 vs. 210 Unify formatting w/v
209 0.5 M NaCl (without pH adjustment?), see 210
216 [MAJOR] What method was used for protein concentration assessment? Were the same amounts loaded per lane in PHA/PHAS gels?
223 [MAJOR] Table S10 lists more than 470 specimens analyzed. Why the discordance? Why are some of these samples colored yellow - to a varying extent? Table 1 gives yet different values of N (430; 314) depending on the dataset subdivision, with no explanation to the discordances.
237-244 [MAJOR] Was this analyzed at per-accession or at per-landrace manner? This missing information is important because of varying number of accessions per landrace.
255 rare {define: what allele frequency cutoff was used?}
267 [MAJOR] Missing: No bootstrap values given at any trees in this study.
270 [MAJOR] Please define what genetic distance algorithm was used to generate this matrix. Several options are available in GenAlEx.
287 [MAJOR] For all statistical tests (Kruskal-Wallis, Wilcoxon, AMOVA), please append for number of permutations per each analysis, and add significance of results in the respective data items presented.
336 associated to -> with
386 (Figure S6A) [MAJOR] The relevant data is not presented there, or it is mislabeled.
428 [MAJOR] Please re-generate Figure S8 to a much higher resolution; in the current version, no labels are legible at any zoom level.
486-488 [MAJOR] Colors in the Figure do not match the legend.
496 Please add something to the effect of "from the SSR analyses", to indicate what data was underlying this visualization.
519 [MAJOR] Colors in this figure do not match the legend.
531 "group membership" was "genetic cluster membership" meant here? Several grouping strategies were used throughout.
531 [MAJOR] group membership and growth habit / PHAS pattern {This needs to be followed by two results of these separate analyses; only one is presented}.
545-551 [MAJOR] What big-picture conclusion can be drawn from comparisons of AMOVA results at per-landrace vs. 2 or 3 PHAS groups?
610 the first three widespread (confusing; was "the top three most prevalent" or similar meant here?}
636 each other {Support with a relevant citation}
655 Genetic Molecular(?) - see 656
683 [MAJOR] "considered as mixtures of genetically distinct lines" Please discuss this more, in the context of your SSR results (VERY high Ho) and the plant's self-pollinating character.
731 "almost similar" Please rephrase.
803 vs. 808 Finally, <-> Furthermore, {Consider exchanging the places}
864 [MAJOR] K= {missing}. {Explain the yellow markings}
Response to Reviewer Point 14:
If several methods are available (which is a brand new information for me regarding generation of rooted trees), please reveal in the manuscript text (M&M) which method was chosen and why.
Response to Reviewer Point 15:
One point that remains unanswered in the manuscript is, How was the clades separation established.
Author Response
Response to Reviewer 2 Comments
Point 1: 56,299,316,338,377,441,487,726 while -> whereas
Response 1: Following the suggestion from the Reviewer, we changed "while" with "whereas” in all sentences indicated.
Point 2: 129 Please unify the use of serial comma (Oxford comma) throughout.
Response 2: Following the Reviewer's suggestion, we unified the use of the serial comma in the indicated sentence and verified its use throughout the text of the manuscript.
Point 3: 146 points of view -> traits(?)
Response 3: Following the Reviewer’s suggestion, we corrected the sentence as follow: “In the present study, a large P. vulgaris collection from the Lazio region (Central Italy) was characterized for morphological traits, and biochemical and molecular markers”.
Point 4: 149-150 EC PDO abbreviation unknown.
Response 4: We thank the Reviewer for the suggestion. We have explained the abbreviation in the line 150 of the present revised version.
Point 5: 163 whole -> entire(?)
Response 5: Done. We replaced “whole” with “entire” in line 164 of the present revised version.
Point 6: 199-200 To keep things more legible, please refrain from juggling the terms or using synonyms. Is "indeterminate" type II? How does that compare to bushy / climbing used later on (384-385)? This paper is complicated enough that adding extra layers only confounds the possible confusion.
Response 6: We thank the Reviewer for the comment. Regardind growth habit the International Board for Plant Genetic Resources (IBPGR) classifies common bean plants into six classes or types, of which five can be considered as different forms of the indeterminate growth habit: type I determinate bush; II indeterminate bush with erect branches; III indeterminate bush with prostrate branches; IV indeterminate with semi-climbing main stem and branches; V indeterminate with moderate climbing ability and pods distributed evenly up the plant; VI indeterminate with aggressive climbing ability and pods mainly on the upper nodes of the plant. In order to simplify the classification of plants on the basis of growth habit, we decided to include growth types II to VI in one class (inderminate growth habit), in accordance with Singh 1982 [47], as stated in MM. For this reason, we reported that determinate growth plants can only be of "type I growth habit," while indeterminate growth plants cannot be classified as belonging to the "type II growth habit”. However, to avoid confusion in interpreting the meaning of “plants with climbing hability” or “bush plants”, we replaced these terms with "plants with indeterminate growth habit" and "determinate growth plants" (see lines 391-393 of the present revised version).
Point 7: 209 vs. 210 Unify formatting w/v
Response 7: Done.
Point 8: 209 0.5 M NaCl (without pH adjustment?), see 210
Response 8: We performed the PHAS and PHA extraction according to the protocol used from Limongelli et al. [Reference N. 48 in the list of references], in which no pH adjustment was used for the PHAS extraction.
Point 9: 216 [MAJOR] What method was used for protein concentration assessment? Were the same amounts loaded per lane in PHA/PHAS gels?
Response 9: We thank the reviewer for the questions. The analysis performed in SDS PAGE is not a quantitative but a qualitative analysis that allows the detection of the different patterns of the two protein fractions (PHAS and PHA) based on two different extraction methods as described in MM. It is obvious that to accurately detect the different patterns of the two protein fractions, the same amount of extract was loaded into the gel for each sample analyzed, which has now been indicated in the line 214 of the present revised version.
Point 10: 223 [MAJOR] Table S10 lists more than 470 specimens analyzed. Why the discordance? Why are some of these samples colored yellow - to a varying extent? Table 1 gives yet different values of N (430; 314) depending on the dataset subdivision, with no explanation to the discordances.
Response 10: We thank the Reviewer for the above comments. First, the number of genotypes in Table S10 turned out to be a total of 472 and not 460 because for each of the four reference genotypes MIDAS, JALO, BAT, and G12873 their SSR profile had been repeated four times to obtain the respective Structure bar plots of average proportion of membership for K=2 and K=3 clusters used in Figure 5. This for uniformity with the Structure bar plots of average proportion of membership for K=2 and K=3 clusters of the four genotypes of each of the 114 accessions reported in the same Figure. However, this potential misunderstanding has been corrected in the current version of Table S10, where for each of the four reference genotypes a single value is reported for the Posterior membership coefficient following the STRUCTURE analysis with K = 2 and K = 3. Now the revised version of Table S10 contains values for 460 genotypes (114 x 4 + 4 references).
Genotypes that can be considered to be of mixed origin at K=2 and K=3 as a result of Structure analysis have been highlighted in yellow in Table S10. To better clarify this we have included the following sentence in the Table caption: “In yellow were labeled the genotypes of the accessions that were considered to have admixture profiles in the STRUCTURE analysis (Posterior membership coefficient Q<0.8) for k=2 and k=3”.
Compared to the 456 individuals related to the 114 accessions of the 66 landraces, the number of individuals reported in Table 1 by classifying the landraces into two or three groups is lower because, as explained in MM (lines 288-290 of of the present revised version), those found to be of “mixed origin” and those that did not show correspondence between PHAS type and cluster membership in STRUCTURE were excluded from the genetic diversity analysis. Without further burdening the text of the manuscript, it is our opinion that with the minor changes made, the number of individuals reported in Table 1 classifying the landraces into two and three groups, 430 and 314, respectively, can be easily deduced from the sentences given in lines 514-524 and 544-552, respectively. For instance at K=2, taking into account that six accessions belonging to five landraces could be considered as putative hybrids between the Andean and Mesoamerican gene pools, as well as two genotypes of one of the four accessions of the landrace "Verdolino," the total genotypes of “mixed origin” is 6 x 4 +2= 26, which subtracted from the 456 genotypes is just equal to 430, as reported in the Table. Similarly at K=3, considering that 35 accessions belonging to 24 landraces could be derived from hybridization among the three groups, as well as the two genotypes of one of the four accessions of the landrace "Verdolino," the total number of genotypes of mixed origin is equal to 35 x 4 +2= 142, which subtracted from the 456 genotypes is exactly equal to 314, as reported in the Table.
Point 11: 237-244 [MAJOR] Was this analyzed at per-accession or at per-landrace manner? This missing information is important because of varying number of accessions per landrace.
Response 11: As clearly indicated in the captions of Figures S2-S6 and in the text of the results section these analyses were performed for each landrace and not for accession. However, to take into account the reviewer's suggestion we have specified this in the indicated text (see lines 238 and 246 of the present revised version).
Point 12: 255 rare {define: what allele frequency cutoff was used?}
Response 12: We thank the Reviewer for the suggestion. The number of rare alleles was determined at the 5% (allele frequency < 0.05) criterion. This information is added in line 257 of the present revised version.
Point 13: 267 [MAJOR] Missing: No bootstrap values given at any trees in this study.
Response 13: We tank the reviewer for the above comment. As described in the subheading 2.6 “Data analyses” of the MM, the distance-based dendrograms with Nei’s genetic distance and UPGMA algorithm were developed using MEGAX. This means that we imported into MEGAX a file of Nei's genetic distances using UPGMA algorithm obtained from Powermarker to visualize the relative trees. Using this procedure, the bootstrapping analysis cannot be performed directly in MEGAX. For this reason, as reported in the the subheading 2.6 “Data analyses” of MM, the reliability of the trees’ topology was assessed via bootstrapping over 1000 replicates using the PAUP* 4.0 software. In order to not further burden graphically the dendrograms shown in Figures 3, 5, S8, and S9, which already contain numerous labels (PHAS types, the names of landraces or accessions, and the numerical code of seed morphotype), we preferred to not report the bootstrap values determined via PAUP 4.0 software. However, the bootstrap values have been reported directly in the text of the manuscript as data not shown in the description of the main groupings obtained by cluster analysis: M and A; A1 and A2 (see lines 438-440 and lines 445-448 of the present revised version).
Point 14: 270 [MAJOR] Please define what genetic distance algorithm was used to generate this matrix. Several options are available in GenAlEx.
Response 14: We thank the Reviewer for the suggestion. As reported on page 6 of GenAlEx Tutorial 2 [Genetic Distance and Analysis of Molecular Variance (AMOVA)] and on page 61 of GenAlEx Guide, we used Codominant Genotypic Distance, which is is the most important genetic distance option for codominant data, since the matrix generated is used in GenAlEx for subsequent PCoA, Mantel and all Spatial analyses. Since this is the only option for calculating genetic distances for codominant data for the Mantel test in GenAlEx, it is our opinion that it is not necessary to provide this information in the MM section, whereas, according to requests made by the reviewer elsewhere, we have indicated the number of permutations used in that analysis (see lines 276-277 of the present revised version).
Point 15: 287 [MAJOR] For all statistical tests (Kruskal-Wallis, Wilcoxon, AMOVA), please append for number of permutations per each analysis, and add significance of results in the respective data items presented.
Response 15: We thank the Reviewer for above comment, even if we were unable to understand its meaning and implications in full. Indeed, it is surprising for us to indicate the number of permutations for Kruskal-Wallis and Wilcoxon nonparametric tests. The significance of the two tests is indicated in Table 1 by the P value in the comparison of the means of the genetic parameters considered between two and three groups, which in both cases is higher than the significance level alpha = 0.05. If the reviewer deems it appropriate, we can add the same letters for the means of the genetic parameters for the two and three groups, although we believe that reporting the P value of the two tests is sufficient to indicate that there is no significant difference between the means.
Instead, following the reviewer's suggestion we have described in detail how the AMOVA was carried out in GenAlEx6, by also indicating the number of permutations used to test statistically the variance components (see lines 261-263 of the present revised version). The P(Phi) statistic probability level after 999 permutations was also reported in Table S8 and Table 2.
Point 16: 336 associated to -> with
Response 16: Done. We replaced “associated to” with “with” in line 343.
Point 17: 386 (Figure S6A) [MAJOR] The relevant data is not presented there, or it is mislabeled.
Response 17: We thank the Reviewer for above comment, even if we were unable to understand its meaning. If the reviewer refers only to Figure S6A we believe we have clearly illustrated the results highlighted in the figure with the sentences reported in lines 390-394 of the present revised version. Or is the reviewer referring to the results for Figures S6B-D? In this case if the reviewer deems it appropriate we could add the following sentences to comment in detail the data highlighted in Figures S6B-D: “Eight of the nine seed colours were detected in landraces of the C-type (Fig. 4B), with maroon, pale cream to buff, and white being the most common. The T-type landraces were characterized by the absence of brown, pale to dark, yellow to greenish yellow, and green to olive seeds, as well as the prevalence of maroon and white seeds, whereas the S-type landraces had mainly white seeds (Fig. 4B). The landraces with the C- and T-types showed in prevalence larger seeds with oval, cuboid or kidney shapes, while the S-type landraces were generally characterized by the presence of cuboid and medium-small sized seeds (Fig. 4C and 4D)”.
We therefore invite the reviewer to let us know whether our interpretation is correct and thus whether we should insert the above sentences.
Point 18: 428 [MAJOR] Please re-generate Figure S8 to a much higher resolution; in the current version, no labels are legible at any zoom level.
Response 18: The Figure S8 is inserted in the supplementary materials also as single TIFF image to improve the reading of the labels.
Point 19: 486-488 [MAJOR] Colors in the Figure do not match the legend.
Response 19: We corrected the colors for the Figure 4 in the lines 494-496 of of the present revised version.
Point 20: 496 Please add something to the effect of "from the SSR analyses", to indicate what data was underlying this visualization.
Response 20: Thank you for your comment. The caption has been improved as required in the lines 504-507 of the present revised version.
Point 21: 519 [MAJOR] Colors in this figure do not match the legend.
Response 21: We corrected the legend for the colors of the Figure 5 in the line 531-532 of the present revised version.
Point 22: 531 "group membership" was "genetic cluster membership" meant here? Several grouping strategies were used throughout.
Response 22: Done. We replaced "group membership" with "genetic cluster membership" in line 543.
Point 23: 531 [MAJOR] group membership and growth habit / PHAS pattern {This needs to be followed by two results of these separate analyses; only one is presented}.
Response 23: We thank the Reviewer for the comment, and following his suggestion we reported in the text the results of the two different analyses: association between genetic cluster membership and growth habit and association between genetic cluster membership and PHAS pattern (see lines 547-548 of the present revised version). We apologise for this omission, which probably caused confusion in the reviewer's interpretation of the results.
Point 24: 545-551 [MAJOR] What big-picture conclusion can be drawn from comparisons of AMOVA results at per-landrace vs. 2 or 3 PHAS groups?
Response 24: We thank the Reviewer for the above comment. It is our opinion that the results obtained by AMOVA analysis by classifying the landraces into two and three groups could arise partly from the observed intra-landrace variability, but mainly from the presence of gene flow between the landraces. We have taken this into account in the present revised version of the manuscript by inserting a sentence in both the results and discussion sections (see lines 562-564 and lines 725-727).
Point 25: 610 the first three widespread (confusing; was "the top three most prevalent" or similar meant here?}
Response 25: Done. We replaced “the first three widespread” with “the top three most prevalent” in line 624.
Point 26: 636 each other {Support with a relevant citation}
Response 26: Following the suggestion of the Reviewer we added a reference in line 650 of the present revised version [No. 69 in the Reference list (Lioi, 1999)].
Point 27: 655 Genetic Molecular(?) - see 656
Response 27: We replaced “Genetic” with “Molecular” in line 669.
Point 28: 683 [MAJOR] "considered as mixtures of genetically distinct lines" Please discuss this more, in the context of your SSR results (VERY high Ho) and the plant's self-pollinating character.
Response 28: We thank the Reviewer for above comment, even if we were unable to understand its meaning and implications in full. The sentence reported by the Reviewer indicates that many of the landraces analyzed show some level of variability within them, and this was possible because four individuals per accession, and, in some cases several accessions per landrace, were analyzed. These results are important, on the one hand, to point out that analyses conducted considering a single individual per landrace, as usually carried out, are not sufficient to reveal the full genetic diversity within them, and, on the other hand, to encourage in situ on-farm conservation of landraces to avoid loss of genetic variability, as discussed in the part below the indicated sentence. Actually we do not understand the need to “discuss this sentence in detail in the context of your SSR results (VERY high Ho) and the plant's self-pollinating character”. First, we detected a very low value of Ho, as exptected in a mainly pollinated species. Second, although few studies have been focused on the analysis of intra-landrace variability in common bean, heterogeneity within landrace accessions is frequently reported in genetic studies from other autogamous species, such as common and durum wheats, as a result of the presence of a set of distinct pure lines, as found in this work for the bean landraces analyzed.
Point 29: 731 "almost similar" Please rephrase.
Response 29: Following the suggestion of the Reviewer, the sentence was chenged as follow: “For instance, in Africa as a whole, the frequencies of the two gene pools are roughly equivalent…” (lines 746 to 748).
Point 30: 803 vs. 808 Finally, <-> Furthermore, {Consider exchanging the places}
Response 30: Done. According with the Reviewer suggestion’ we exchanged the place of the two adverbs.
Point 31: 864 [MAJOR] K= {missing}. {Explain the yellow markings}
Response 31: We thank the Reviewer for the suggestion and added the missing K value (=3) and the legend of yellow marking in the table S10 (Lines 880-882).
Point 32: Response to Reviewer Point 14:
If several methods are available (which is a brand new information for me regarding generation of rooted trees), please reveal in the manuscript text (M&M) which method was chosen and why.
Response 32: We apologize to the Reviewer for the misunderstanding that arose with the response provided in point 14 of the first review. As mentioned above (answer to point 13) to obtain the dendrograms and their optimal visualization, we used the MEGAX software. In particular, we imported into MEGAX a file of Nei's genetic distances using UPGMA algorithm obtained by the software Powermarker. Since UPGMA method assumes that the rate of evolution has remained constant throughout the evolutionary history of the included individuals, the produced tree is a rooted tree. In MEGAX a rooted tree is one in which the root of the phylogenetic tree is determined by using the mid-point rooting or by indicating a single genotype or a group of genotypes as out-group. The trees reported in the manuscript are obtained by using the mid-point rooting method, in which the root of an unrooted tree is placed at the mid-point of the longest distance between two individuals in a tree. In our case it is difficult to identify a landrace or a group of landraces to be used as an outgroup, as our aim is to study only the relationships among the landraces analyzed and eventually to individuate their appartentenence to the Mesoamerican and Andean gene pools. Indeed, in our study, we analyzed genetic relationships between the considered genotypes, accessions and landraces, including BAT and G12873 of Mesoamerican origin, and MIDAS and JALO of Andean as reference genotypes. Since numerous studies used MEGAX software for visualizing the dendrograms of genetic distances without reporting all the details on obtaining rooted and unrooted trees, in order not to further burden the text of the manuscript we believe that it is not necessary to report the method used to obtain the rooted trees.
In the answer given in the first review we didn't want to be offensive to the reviewer, but this is the reality, as there are several methods to root phylogenetic trees, which include: outgroup, midpoint rooting, molecular clock rooting, and Bayesian molecular clock rooting (see for instance Kinene T, Wainaina J, Maina S, Boykin LM. Rooting Trees, Methods for. Encyclopedia of Evolutionary Biology. 2016:489–93. doi: 10.1016/B978-0-12-800049-6.00215-8. Epub 2016 Apr 21. PMC ID: PMC7149615).
Point 33: Response to Reviewer Point 15:
One point that remains unanswered in the manuscript is, How was the clades separation established.
Response 33: As reported in our previous responses, Figure 4 shows the Wright’s population pairwise fixation index (Fst) values. The clustering has been developed on the basis of Fst bins using the UPGMA method. The caption of Figure 4 has been improved.

Round 3
Reviewer 2 Report
Dear Authors,
At this stage I have only minor comments and suggestions. After these are incorporated, I suggest your manuscript for publication.
102 (It is recognized that) {Can be safely deleted with no change in meaning of this sentence}
104 center{s}
113-114 (intentional or unconscious) {Weird phrasing. Consider "deliberate or unintentional"}
116 (a great number of landraces) {Support this claim with a citation - for instance a national or EU registry of landraces}
251 (and few more instances) analyse - please unify throughout for the American spelling.
384 (1-6) {patterns 1 to 6}
393 {Consider, for symmetry with 391, "plants with determinate growth habit"}
Figures - Unify the presence of frames
421 (the greater number) {the comparative requires a follow-up: "greater... than..." An alternative is to state "the highest number of private alleles"}
430 (an accession) {"one accession"?}
484 Please unify the serial commas throughout - here and in several other instances. In Response to Reviewers Authors claimed it was corrected.
492 (in agreement to) {in agreement with}
492 (most of) {majority of}
510 (and several other places) (germplasm collection) {Isn't this a pleonasm? Consider retaining only one}
530 (Consider retaining only one term throughout for "gene pools" 512, "clusters", "genetic group", and others - juggling the terms for the same thing is not helping readability of this paper in my opinion)
534 (Mesoamerica{n})
541 (accounted... of) {Weird phrasing; perhaps "consisted... of"?}
567 (while) {whereas}
568 (Mesoamerica{n})
570 (similar results {to those})
640 {Here and several other places: Unify the spacing between sentences}
650-652 (than predicted random assortment) {This sentence reads awkward. Consider rephrasing}
659 (pattern {of?})
685 (and even more private alleles) {reads awkward; consider rephrasing}
698 (would not been enough) {Incorrect grammar}
699 (tree(s})
739 (the most probable number) {Elsewhere referred to as "optimum number" Please unify throughout}
749 (mostly) {in major part?}
794 (lower to) {lower than}
810 vs. 815 (the) Lazio region {Consider unifying}
Author Response
Response to Reviewer 2 Comments
Point 1: 102 (It is recognized that) {Can be safely deleted with no change in meaning of this sentence}
Response 1: Following the Reviewer’ suggestion we corrected the sentence in line 102.
Point 2: 104 center{s}
Response 2: Done.
Point 3: 113-114 (intentional or unconscious) {Weird phrasing. Consider "deliberate or unintentional"}
Response 3: Following the Reviewer’ suggestion we corrected the sentence in lines 113-114.
Point 4: 116 (a great number of landraces) {Support this claim with a citation - for instance a national or EU registry of landraces}
Response 4: Following the Reviewer’ suggestion we added the Reference n.34 in line 116.
Point 5: 251 (and few more instances) analyse - please unify throughout for the American spelling.
Response 5: We thank the Reviewer for the suggestion. Because “analyse” and “analyze” are two variants of the same verb, and we mainly used in the text “analyse”, we unified throughout the manuscript with the British spelling.
Point 6: 384 (1-6) {patterns 1 to 6}
Response 6: Following the Reviewer’ suggestion we corrected the sentence in line 385.
Point 7: 393 {Consider, for symmetry with 391, "plants with determinate growth habit"}
Response 7: Following the Reviewer’ suggestion we corrected the sentence in line 394.
Point 8: Figures - Unify the presence of frames
Response 8: Done.
Point 9: 421 (the greater number) {the comparative requires a follow-up: "greater... than..." An alternative is to state "the highest number of private alleles"}
Response 9: Following the Reviewer’ suggestion we corrected the sentence in line 422.
Point 10: 430 (an accession) {"one accession"?}
Response 10: Following the Reviewer’ suggestion we corrected the sentence in line 431.
Point 11: 484 Please unify the serial commas throughout - here and in several other instances. In Response to Reviewers Authors claimed it was corrected.
Response 11: Done.
Point 12: 492 (in agreement to) {in agreement with}
Response 12: Following the Reviewer’ suggestion we corrected the sentence in line 493.
Point 13: 492 (most of) {majority of}
Response 13: Following the Reviewer’ suggestion we corrected the sentence in lines 493-494.
Point 14: 510 (and several other places) (germplasm collection) {Isn't this a pleonasm? Consider retaining only one}
Response 14: Done.
Point 15: 530 (Consider retaining only one term throughout for "gene pools" 512, "clusters", "genetic group", and others - juggling the terms for the same thing is not helping readability of this paper in my opinion)
Response 15: We thank the Reviewer for the suggestion. We left the term “gene pools” when in the text the Mesoamerican and Andean origins of the accessions were mentioned. On the other hand we unified “genetic groups” with “clusters” as suggested.
Point 16: 534 (Mesoamerica{n})
Response 16: Done.
Point 17: 541 (accounted... of) {Weird phrasing; perhaps "consisted... of"?}
Response 17: Following the Reviewer’ suggestion we corrected the sentence in line 542.
Point 18: 567 (while) {whereas}
Response 18: Following the Reviewer’ suggestion we corrected the sentence in line 569.
Point 19: 568 (Mesoamerica{n})
Response 19: Done.
Point 20: 570 (similar results {to those})
Response 20: Following the Reviewer’ suggestion we corrected the sentence in line 572.
Point 21: 640 {Here and several other places: Unify the spacing between sentences}
Response 21: Done.
Point 22: 650-652 (than predicted random assortment) {This sentence reads awkward. Consider rephrasing}
Response 22: We thank the Reviewer for the suggestion. We slightly changed line 654 because the meaning of the sentence is clear in our opinion.
Point 23: 659 (pattern {of?})
Response 23: Done.
Point 24: 685 (and even more private alleles) {reads awkward; consider rephrasing}
Response 24: Following the Reviewer’ suggestion we corrected the sentence in line 687.
Point 25: 698 (would not been enough) {Incorrect grammar}
Response 25: Following the Reviewer’ suggestion we corrected the sentence in line 700.
Point 26: 699 (tree(s})
Response 26: Done.
Point 27: 739 (the most probable number) {Elsewhere referred to as "optimum number" Please unify throughout}
Response 27: Done.
Point 28: 749 (mostly) {in major part?}
Response 28: Following the Reviewer’ suggestion we corrected the sentence in line 751.
Point 29: 794 (lower to) {lower than}
Response 29: Following the Reviewer’ suggestion we corrected the sentence in line 796.
Point 30: 810 vs. 815 (the) Lazio region {Consider unifying}
Response 30: Done.
